

# Bulk density and its connection to other microphysical properties of snow as observed in Southern Finland

Jussi Tiira[1], Dmitri N. Moisseev[1, 2], Annakaisa von Lerber[2, 3], Davide Ori[1, 4], Ali Tokay[5, 6], Larry F. Bliven[7], and Walter Petersen[8]

[1]Department of Physics, University of Helsinki, Helsinki, Finland
[2]Finnish Meteorological Institute, Helsinki, Finland
[3]School of Electrical Engineering, Aalto University, Finland
[4]Department of Biological Geological and Environmental Sciences and Department of Physics and Astronomy, University of Bologna, Bologna, Italy
[5]Joint Center for Earth Systems Technology, University of Maryland, Baltimore County, Baltimore
[6]NASA Goddard Space Flight Center, Greenbelt, MD
[7]NASA GSFC/Wallops Flight Facility, Wallops Island, VA
[8]NASA-MSFC Earth Science Office, National Space Science and Technology Center, Huntsville, AL

*Correspondence to:* Jussi Tiira (jussi.tiira@helsinki.fi)

**Abstract.** In this study measurements collected during winters 2013/2014 and 2014/2015 at the University of Helsinki measurement station in Hyytiälä are used to investigate connections between snow bulk density, particle fall velocity and parameters of the particle size distribution (PSD). The bulk density of snow is derived from measurements of particle fall velocity and PSD, provided by a particle video imager, and weighing gauge measurements of precipitation rate. Validity of the retrieved density

5    values is checked against snow depth measurements. A relation retrieved for bulk density and median volume diameter is in general agreement with previous studies, but observed to vary significantly from one winter to the other. From these observations, characteristic mass-dimensional relations of snow are retrieved. For snow rates more than $0.2 \, \mathrm{mm \, h^{-1}}$, a correlation between the intercept parameter of normalized gamma PSD and median volume diameter was observed .

## 1 Introduction

10   Due to a variety of ice particle types and shapes, representation of winter precipitation in models (Woods et al., 2007; Morrison and Milbrandt, 2015) and in ground, airborne, and satellite remote sensing retrievals (Sekhon and Srivastava, 1970; Matrosov, 1997; Wood et al., 2013) is a topic of continuous interest. Both models and retrieval algorithms rely on a prior knowledge of snowflake mass, shape and fall velocity, which are typically expressed as functions of a characteristic particle size (Pruppacher and Klett, 1996). Furthermore, information on possible particle size distributions (PSDs) is also required. Even though some

15   of the microphysical properties of ice particles are not independent, e.g. fall velocity can be computed from particle mass and shape (Böhm, 1989; Khvorostyanov and Curry, 2005; Mitchell and Heymsfield, 2005; Heymsfield and Westbrook, 2010), the remaining degrees of freedom are rather numerous.





Historically, measurements of snowflake properties have been carried out on particle-by-particle basis (e.g., Magono and Nakamura, 1965; Locatelli and Hobbs, 1974; Mitchell, 1996). While we may still regard such measurements as the more precise and detailed, due to the sheer amount of time needed for such experiments and corresponding data analysis, these studies are limited to a relatively small number of observed ice particles. After introduction of robust optical instruments

capable of measuring particle size, shape and in some cases fall velocity, e.g. 2D-video disdrometer (2DVD, Hanesch, 1999; Schönhuber et al., 2007), particle size velocity (Parsivel) laser-optical disdrometer (Löffler-Mang and Joss, 2000; Löffler-Mang and Blahak, 2001), hydrometeor velocity size detector (HSVD, Barthazy et al., 2004), snow video imager (SVI, Newman et al., 2009) and multi-angle snowflake camera (MASC, Garrett and Yuter, 2014), continuous recording of ice particle properties became possible. It should be noted, in comparison to surface based observations, aircraft measurements have a much longer

history in determining ice particle microphysical properties, and were carried out in different types of clouds and climate regimes (Pruppacher and Klett, 1996). A typical limitation of automatic observations of ice particle properties, however, is that only a subset of needed parameters is directly measured.

By combining optical disdrometer observations with other measurements, i.e. by radar or precipitation gauge, physical properties such as snow density can be derived. Huang et al. (2010) have used a C-band weather radar observations of equivalent

reflectivity factor, $Z_e$, in combination with a 2DVD to derive snow density-dimensional relation and to infer more consistent $Z_e$ – snowfall rate, SR, relations. Brandes et al. (2007), hereafter referred to as B07, used a combination of a weighing gauge and a 2DVD to derive bulk density-median volume diameter relations and to document relations between PSD parameters for Colorado winter storms. Their approach is similar to the one used by Heymsfield et al. (2004) who have combined aircraft PSD and ice water content observations to derive bulk density and average mass-dimensional relations. Albeit using slightly

different definitions, both B07 and Heymsfield et al. (2004) derive effective ice densities for ice particle populations, but there is a difference in terminology. We follow terminology of B07 and refer to the derived densities as bulk densities whereas Heymsfield et al. (2004) talk about effective densities of particle populations. Another method for snow density retrieval is based on solving aerodynamic equations to derive particle mass from observed fall velocity and particle effective projected area as proposed by Böhm (1989) and applied by Hanesch (1999) and more recently by Szyrmer and Zawadzki (2010) and

Huang et al. (2015).

This paper documents connection between bulk density and other microphysical properties of snow as observed in Southern Finland. From the estimated bulk density, average mass-dimensional relations characteristic to studied snowfall events are defined. In order to derive bulk snow density, a method proposed by B07 was used. However, instead of a 2DVD, a new generation of the SVI is employed. It is shown that, despite simpler construction compared to the 2DVD, this instrument's data

is suitable for such studies.



## 2 Measurements

### 2.1 Measurement setup

Measurements were made at the University of Helsinki Hyytiälä Forestry Field Station, Finland (N61°50′37″, E24°17′16″) during the Biogenic Aerosols Effects on clouds and Climate (BAECC) field campaign (Petäjä et al., 2016) and during the consecutive winter of 2014/15. BAECC was a joint experiment between the University of Helsinki, the Finnish Meteorological Institute and the United States Department of Energy Atmospheric Radiation Measurement (ARM) program. From 1 February through 12 September 2014 the second ARM Mobile Facility (AMF2) was deployed to the measurement site. The measurement setup was designed for snowfall intensive observation period (IOP) of BAECC, called BAECC Snowfall Experiment (SNEX), which was undertaken from 1 February though 30 April 2014 and focused on measurements of snow microphysics. However, in order to extend the dataset, the measurements were continued upon completion of BAECC. In total 23 snowfall cases from winters 2013/14 and 2014/15, where a number of events extended over a couple of days, were used in this study as summarized in Table 1. The snowfall cases were selected based on measurements of liquid water equivalent (LWE) precipitation accumulation by a weighing gauge, snow depth using a laser sensor, and temperature measured by the automatic weather station of FMI located 500 m distance from the measurement site. Only precipitation cases, where temperature was below or equal to 0°C were chosen, and when occasionally the temperature during the event rose above 0°C, the data was omitted.

The experiments in both winters were organized in collaboration with the National Aeronautics and Space Administration (NASA) Global Precipitation Measurement (GPM) Mission ground validation (GV) program. The surface precipitation measurements are carried out using a number of collocated instruments, such as NASA Particle Imaging Package (PIP), two OTT Pluvio[2] weighing gauges, two Parsivel[2] laser disdrometers (Tokay et al., 2014), a 2DVD and a laser snow depth sensor by Jenoptik. To minimize effects of wind, a Double-Fence Intercomparison Reference (DFIR) wind protection (Rasmussen et al., 2012) was build on site as shown in Fig. 1 and discussed in more detail in Petäjä et al. (2016). Inside of the DFIR, the 2DVD, one of the OTT Pluvio[2]s and one of the Parsivel[2] disdrometers were placed. In addition to the precipitation sensors, 3D-anemometers were deployed. The wind measurements were carried out at the heights of precipitation instrument sampling volumes. In this study data from the NASA PIP disdrometer and both OTT Pluvio[2] gauges are used.

### 2.2 Particle Imaging Package (PIP)

The NASA Particle Imaging Package is the new generation of the SVI. The PIP, like the SVI, consists of a halogen lamp and a charge-coupled device (CCD) full frame camera with sensor resolution of $640 \times 480$ pixels. The main differences between PIP and SVI are the camera and improved software. The camera is now capable of imaging with a frame rate of 380 frames per second, enabling measurements of particle fall velocities. The distance between the lamp and the camera lens is approximately 2 m. The lens focus is set at 1.3 m, where the field of view (FOV) is $64 \times 48$ mm, and the image resolution thereby $0.1 \times 0.1$ mm. The main advantage of PIP, as well of SVI, over other disdrometers is the open particle catch volume, which minimizes effect of wind on quantitative precipitation measurements (Newman et al., 2009).



The instrument records shadows of particles as they fall through the observation volume. Given the camera frame rate, multiple images of a particle are recorded and used to estimate its fall velocity. The depth of field (DOF) is determined by the processing software either rejecting or not detecting particles that are out of focus. Thus, the observation volume is defined by the FOV and the DOF. The expected particle size error due to the blurring effect is 18 % (Newman et al., 2009). From

the recorded particle images a number of parameters describing particle geometrical properties are calculated with National Instruments IMAQ - software. The measured diameter is given as the equivalent disk diameter, which is the diameter of a disk with the same area as the area of a particle image. Other parameters, such as particle orientation, equivalent ellipse major and minor axes are also recorded.

### 2.3   Weighing gauges and anemometers

The measurement setup includes two OTT Pluvio$^2$ weighing gauges. The one inside of the DFIR has an orifice of 200 cm$^2$ and the one outside an 400 cm$^2$ orifice. There are differences in wind shielding as well. The Pluvio$^2$ 200 is equipped with a Tretyakov wind shield and the Pluvio$^2$ 400 with a combination of Tretyakov and Alter wind shields, as seen in the forefront in Fig. 1.

   The gauges output several products of precipitation rate and accumulation. In this study, a non-real-time accumulation

product is used as it is filtered for various sources of errors such as changes in the bucket mass due to evaporation, and as such should yield the most precise precipitation rate estimate among the output products. Because of the filtering, there is a 5 min delay in the recorded time series, which needs to be taken into account when comparing to other instruments. The precipitation accumulation values are recorded with a resolution of 0.001 mm, but non-real-time accumulation is output with a resolution of 0.05 mm.

The 3D - anemometer manufactured by Gill is located approximately at the height of the PIP on the field, respectively. The wind parameters, horizontal and vertical speed and horizontal direction, of Gill anemometer are measured every 10 s and averaged over 60 s. The mean and maximum of the 60 second wind speed averages and the mean wind direction for each event are given in the Table 1.

### 2.4   Snow depth sensor

The laser snow depth sensor, Jenoptik SHM30, is located on the measurement field, next to Pluvio$^2$ 400. It is an optic sensor, which measures the snow depth by comparing signal phase information of the modulated visible laser light. It is a point measurement, hence the piling of wind driven snow or random branches and leaves drifting on the snow pack can cause misreadings. To reduce this we have sheltered the measurement spot with a small wind fence and the instrument structure excluding the measurement pole is buried under the ground to prevent the piling of snow. The data is recorded every minute.





## 3  Retrievals of bulk density, velocity-dimensional relations and PSD

Observations from the PIP and one of the weighing gauges are combined to retrieve bulk snow density. Typically the gauge located inside of the DFIR, the Pluvio$^2$ 200, is used for this retrieval. On a couple of days this gauge was not operational and data from the Pluvio$^2$ 400 outside was used instead. These dates are marked in Table 1 with asterisks in the LWE precipitation

rate column. As seen in the Table 1 the differences in accumulated LWE recorded by the two Pluvio$^2$s are small, largest being 15%. Pluvio$^2$ 200 inside the DFIR is typically measuring higher accumulations, which is expected because of the smaller orifice, but there is no clear indication that this would be depended on wind speed. However, the difference seems to increase in respect to certain wind directions. There are two openings from the measurement field, one to a road crossing (approx. 130°) and the other to small field (approx. 180°). If the wind is blowing from these directions the difference between the two gauges

seem to increase.

The retrieval procedure is described below and is similar to the one presented by B07, but with notable modifications. Prior to retrieval of snow bulk density, PSD and velocity-dimensional relations are estimated. It was found, however, that the density retrieval is highly sensitive to the integration time. To minimize this, a variable integration time determined by the precipitation accumulation is used. The same integration time was applied to compute PSD parameters and $v$-$D$ relations.

### 3.1  Particle size distribution (PSD)

The PSDs are calculated from the PIP records of particles that fell through the observation volume. The distributions are defined with respect to equivalent area diameter, which is different from the apparent diameter of the 2DVD and maximum particle dimensions used in other studies (e.g., Heymsfield et al., 2004). Wood et al. (2013) studied differences between diameter definitions and found that the diameter recorded by SVI is approximately 0.82 of maximum particle dimension. We performed

a similar study by examining mean dimensions of a rotated spheroid on a single projection, and found that the PIP diameter is roughly equal to 0.92 of a volume equivalent diameter, i.e. the diameter for which the particle volume $V(D) = \frac{\pi}{6}D^3$. This conversion factor is the mean value for spheroidal particles with axis ratio of 0.6 (Korolev and Isaac, 2003; Matrosov et al., 2005b), and orientation defined by Gaussian distribution of canting angles with the standard deviation of 9° (Matrosov et al., 2005a) and uniform distribution of azimuth angles. The conversion factor is used in our study and all the results are presented

using this volume equivalent diameter proxy.

Prior to calculations of PSD parameters, recorded PSD data is filtered to remove spurious observations of large particles. Following the procedure described in (Leinonen et al., 2012) records of large particles were ignored if there was a gap of more than three consecutive PSD diameter bins. The bin size was set to 0.25 mm during the BAECC experiment and it was reduced to 0.2 mm for the winter 2014/2015. The PIP resolution is 0.1 mm and the minimum detectable particle diameter is

approximately 0.3 mm (Newman et al., 2009). The smallest diameter bin used in calculations is 0.25 mm to 0.5 mm during BAECC and 0.2 mm to 0.4 mm in the following winter.



The PSD parameters were calculated using method of moments and assuming that PSD follows gamma functional form. The normalized gamma distribution $N(D)$ in $\mathrm{mm}^{-1}\mathrm{m}^{-3}$ was adopted following Testud et al. (2001); Bringi and Chandrasekar (2001); Illingworth and Blackman (2002):

$$N(D) = N_w f(\mu) \left(\frac{D}{D_0}\right)^{\mu} \exp(-\Lambda D) \tag{1}$$

$$f(\mu) = \frac{6}{3.67^4} \frac{(3.67+\mu)^{\mu+4}}{\Gamma(\mu+4)} \tag{2}$$

$$\Lambda = \frac{3.67+\mu}{D_0}, \tag{3}$$

with $N_w$ in $\mathrm{mm}^{-1}\mathrm{m}^{-3}$ being the intercept parameter, $D_0$ the median volume diameter in mm, $\Lambda$ the slope parameter in $\mathrm{mm}^{-1}$ and $\mu$ the shape parameter. Using the second, fourth and sixth moments for the non-truncated gamma PSD, $M_2$, $M_4$, and $M_6$, the PSD parameters were estimated as follows:

$$\eta = \frac{M_4^2}{M_6 M_2} \tag{4}$$

$$\mu = \frac{7 - 11\eta - \sqrt{\eta^2 + 14\eta + 1}}{2(\eta-1)} \tag{5}$$

$$\Lambda = \sqrt{\frac{M_2 \Gamma(\mu+5)}{M_4 \Gamma(\mu+3)}} \tag{6}$$

$$D_0 = \frac{3.67+\mu}{\Lambda} \tag{7}$$

### 3.2 Bulk density retrieval

The integration time, $\tau(t)$, of the bulk density retrieval is driven by precipitation measurements of the Pluvio$^2$. The step of the non-real-time accumulation output is 0.05 mm, causing the output interval to be in the order of several minutes even at moderate snow rates. With a short fixed integration time in time scales of minutes or tens of minutes, the produced bulk density estimation would hence be the more unstable, the lower the precipitation rate. Therefore, variable length time intervals driven by the gauge output are used with a selected threshold value of 0.1 mm. This corresponds to a $\tau(t)$ of 6 minutes for a LWE precipitation intensity of $1\,\mathrm{mm\,h^{-1}}$.

As the integration time interval $\tau(t)$ is effectively driven by precipitation intensity, there is less variation in number of particles between intervals, compared to a fixed time interval approach. With the selected accumulation threshold there are typically between $10^3$ and $10^4$ particles within a given integration time interval. On the other hand, with low precipitation intensities, $\tau(t)$ increases up to one hour and retrieved bulk density becomes less representative for the time interval in question. With LWE precipitation rates lower than $0.2\,\mathrm{mm\,h^{-1}}$, the resolution of Pluvio$^2$ LWE measurements is insufficient and calculations of bulk density become overly sensitive to recorded number concentrations. Correspondinly, similar unwanted sensitivity to LWE precipitation accumulation occurs when the number of particles observed by PIP within $\tau(t)$ is less than 800. Therefore, time intervals with precipitation rates or particle counts lower than these thresholds are excluded from our analysis.



Given a population of solid precipitation particles with volume equivalent diameters $D$ over the integration time $\tau(t)$, the liquid equivalent precipitation accumulation in mm is

$$G(t) = \frac{\pi}{6} \times 10^{-6} \frac{\rho}{\rho_w} \int_t^{t+\tau(t)} \int_0^{D_{max}} D^3 v(D,t) N(D,t) \, \mathrm{d}D \mathrm{d}t, \tag{8}$$

where $\rho$ is the volume flux weighted snow bulk density in $\mathrm{g\,cm^{-3}}$, $\rho_w = 1\,\mathrm{g\,cm^{-3}}$ is the density of liquid water, $N(D,t)$ is

mean particle number concentration over the integration time in $\mathrm{mm^{-1}m^{-3}}$ and $v(D,t)$ is particle velocity relation in $\mathrm{m\,s^{-1}}$. From (8) we can calculate snow bulk density for each observation time interval as

$$\rho(t) = \frac{6}{\pi} \times 10^6 \rho_w \frac{G(t)}{\int_t^{t+\tau(t)} \int_0^{D_{max}} D^3 v(D,t) N(D,t) \, \mathrm{d}D \mathrm{d}t}, \tag{9}$$

using liquid equivalent precipitation accumulation $G(t)$ as measured by the Pluvio$^2$ gauge, and retrieving volume flux with fitted $v(D,t)$, and averaged $N(D,t)$ as measured by the PIP. It should be noted, that unlike in the retrieval of PSD parameters,

where gamma PSD was assumed, $\rho$ was retrieved without making any assumptions on the shape of the PSD distribution, and instead, measured PSD are used in the calculations.

The definition of bulk density here is the same as in B07. They determine the densities for 5-minute precipitation volumes derived with a 2DVD disdrometer observations together with precipitation mass measured by a weighing gauge. B07 defined the volume of a single particle by summing coin-shaped sub-volumes together estimated separately for both orthogonal projec-

tions and taking geometrical mean. As the used diameter in our study is the estimated volume-equivalent diameter, our results are comparable to B07. In Heymsfield et al. (2004), the volume of a single particle is defined as a function of circumscribing maximum diameter, and the population mean effective density is determined from ice water content (IWC). The estimated bulk density is volume-weighted and expected to have lower values than the velocity-weighted bulk density. The difference is not generally prominent especially with low-density aggregates, whose velocity-dimensional dependence is weak.

It should be noted that the derived density is inversely proportional to the snow ratio, $R_s$. The snow ratio (Power et al., 1964; Ware et al., 2006) is used by operational weather services to estimate change in snow depth from LWE observations and can be defined as follows:

$$R_s(t) = \frac{\rho_w}{\rho(t)} \tag{10}$$

where $\rho(t)$ is the volume flux weighted bulk density derived as shown in (9). Given this connection, the derived density can be

used to verify the commonly used assumption that 1 mm of LWE accumulation corresponds to 1 cm change in snow depth. In Fig. 2 the combined distribution of estimated snow ratios on temporal scales defined by the gauge accumulation for all the 23 events analyzed in this study is shown. It can be seen that the mean and median values, equal to 10 and 9 respectively, are very close to the commonly assumed value. This supports the validity of the retrieved bulk density values.

One of the major uncertainties in the density retrieval is the assumption about particle volume. In this study we have assumed

that snowflakes are spheroids with axis ratios of 0.6. Given this assumption, a conversion factor relating volume equivalent and observed disc equivalent diameters was defined. It should be noted that the volume flux, defined in the denominator of (9), is



nothing more than rate of snow depth change. Therefore, by comparing the PIP derived and the directly measured snow depths, the validity of the derived bulk density values, and assumption of particle shape, can be checked. In Fig. 3 hourly change in the snow depth measured by the Jenoptik SHD30 is compared to the PIP derived snow depth. It can be seen that the agreement is rather good and there are no systematic differences. This comparison also gives confidence about the validity of the derived

bulk densities.

### 3.3 Velocity-dimensional analysis

For each integration time interval, $v(D) = a_v D^{b_v}$ is computed. The $v(D)$ power law fits to unfiltered data tend to be strongly biased by outliers. To address this problem, Gaussian kernel density estimation (KDE, Silverman, 1986) is used to find the most probable velocity for each diameter bin, and only observations with velocities within half width at half maximum from

the bin peak KDE value are included in calculating the fit. Using the linear least squares method, a fit is performed for the data points in log-log scale to derive a power law relation. Velocity fits retrieved this way are shown for selected integration time intervals of the 18 March 2014 and the 22-23 January 2015 cases in the bottom of Figures 4 and 5, respectively.

It should be noted that the power law model, albeit widely used, may not necessary represent correctly velocities of ice particles over the complete range of diameters (Mitchell and Heymsfield, 2005). In many cases the fit can also be uncertain

either because of narrow PSD or in presence of multiple particle types.

## 4   Results

### 4.1   Case studies

#### 4.1.1   18 March 2014

During the March 18, Finland was covered in a continental polar air mass. In the morning, a warm occluded front associated

with a weak low pressure center approached southern Finland from southwest bringing light snowfall. In the afternoon Hyytiälä was in the warm sector of the frontal system, and the relative humidity dropped halting the snowfall around 12 UTC. Later in the evening there was a one-hour snow shower from a squall line associated with a cold front passing over southern Finland.

Time series of LWE snow rate, snow bulk density and PSD parameters for the March 18 case are shown in Fig. 4. The bottom panels show measured fall velocities for selected integration time intervals representing observations with different

bulk densities. Between the red dotted lines is the region where KDE is higher than half maximum for a given particle size. The fits are applied for data points between these lines. There is considerable scatter in particle fall velocity throughout the case and a bimodal PSD is present momentarily in the morning as can be seen in fall velocity panel Fig. 4a.

During the snow shower in the evening, liquid equivalent precipitation rates on average roughly three times more intense than earlier during the day were recorded, allowing retrievals of bulk density and PSD parameters at high time resolutions.

Strong short time scale variations of $\rho$ and PSD parameters are recorded during this shower. The lowest bulk density value of the case, $0.035 \, \mathrm{g \, cm^{-3}}$, is retrieved for time interval from 16:35 to 16:39, with concurrent $D_0$ value of 5.5 mm and $N_w$ of





roughly 700 $\mathrm{mm}^{-1}\mathrm{m}^{-3}$. The corresponding fall velocity distribution visualized in panel 4b is characterized by low values of velocity fit coefficients $a_v$ and $b_v$. Within the following 20 minutes, $D_0$ decreases down to roughly 2 mm, $N_w$ increases to $2 \times 10^4$ $\mathrm{mm}^{-1}\mathrm{m}^{-3}$, and retrieved values of $\rho$ peak at over 0.2 $\mathrm{g\,cm}^{-3}$ between 16:54 and 16:58, and again from 17:05 to 17:08. Corresponding fall velocity distribution between 16:54 and 16:56, shown in panel 4c, is characterized by substantially higher

values of $a_v$ and $b_v$, than 20 minutes earlier.

### 4.1.2  22-23 January 2015

During 22 January 2015, similarly to the 18 March 2014 event, a warm occluded front associated with a weak low moved northwards over the Gulf of Finland. However, due to a blocking high over north-western Russia, the low and the associated front were sustained over southern Finland for the whole day of January 23rd causing weak continuous precipitation in the

area.

Time series of LWE snow rate, bulk density and PSD parameters for the 22-23 January 2015 case, with velocity-diameter fits from selected time intervals are shown in Fig. 5. The case is characterized by continuous snowfall at LWE precipitation rates lower than 1 $\mathrm{mm\,h}^{-1}$ throughout the case. The velocity distribution for a given time interval has substantially less scatter compared to the 18 March 2014 case. The evolution of $\rho$ and $N_w$, as shown in Fig. 5, show considerable similarities, suggesting

a strong correlation.

The velocity-diameter fits shown represent a low bulk density ($\rho = 0.05\,\mathrm{g\,cm}^{-3}$) time interval 01:03-01:16 (panel 5b) and two intervals 22:30-22:52 and 02:06-02:14 (5a, c) with higher bulk densities 0.10 and 0.12 $\mathrm{g\,cm}^{-3}$, respectively. Notable is the higher modal fall velocities and the absence of particles larger than 3 mm in the high density time intervals compared to the distribution in panel 5b.

### 4.2  v-D and density

In Fig. 6, particle fall velocity versus diameter data points combined from all the cases are divided into three categories according to the bulk density of the time interval during which particles were observed. A least squares fit is applied to observations in each bulk density range using the same procedure as for velocity dimensional fits for integration time intervals, as described in section 3.3. The total number of observed particles is roughly 4,440,000, and for each bulk density category

numbers of particles included in the fitting process (within the red lines in Fig. 6) are approximately 1,140,000, 1,190,000 and 360,000, respectively. The fitted relations for bulk density ranges are

$$v(D) = 0.834 D^{0.217}, \quad 0.0\,\mathrm{g\,cm}^{-3} < \rho \le 0.1\,\mathrm{g\,cm}^{-3}, \tag{11}$$

$$v(D) = 0.895 D^{0.244}, \quad 0.1\,\mathrm{g\,cm}^{-3} < \rho \le 0.2\,\mathrm{g\,cm}^{-3} \text{ and} \tag{12}$$

$$v(D) = 0.906 D^{0.256} \quad \rho \ge 0.2\,\mathrm{g\,cm}^{-3}. \tag{13}$$

The coefficient is increased with density indicating higher fall velocities with more dense particles. There is also a clear increase in the slope of the fitted curve from the lowest bulk density range to the $0.1 \dots 0.2$ $\mathrm{g\,cm}^{-3}$ range indicated by the increase in the power term. With particles in the highest bulk density range the observed size distribution is narrow, hence the



correlation between particle size and fall velocity is weak, and it is difficult to find an unambiguous relation between them. All things considered, the results are in line with the conclusion made by Barthazy and Schefold (2006), that the constant and power terms increase with riming degree.

### 4.3  Connection between PSD parameters and density

From the analysis of PSD parameters and their relations to bulk density we have excluded data points representing integration time intervals where $D_0 < 0.6$ mm, as lower values of median volume diameter would imply that a substantial fraction of particles are too small to be observed with PIP. Applying this restriction, along with minimum thresholds set for particle count and LWE precipitation rate in density retrievals, as described in section 3.2, all in all 101 time intervals were discarded from the total of 1141 intervals of observations, leaving 7173 minutes of snow observations for the analysis.

#### 4.3.1  Density and $D_0$

In Fig. 7, observed distributions of $D_0$ for the three different density regimes are shown. For the low density particles, the maximum $D_0$ value does not seem to exceed $5 \ldots 6$ mm, which is in agreement with observations of snow aggregates presented by Lo and Passarelli (1982). It can also be seen that $D_0$ distribution depends on density. Low density particles are generally larger and vice versa. This dependence of $D_0$ on bulk density is not surprising, given that they are related as was previously shown by B07 and discussed in more detail below.

Relation between snow bulk density and size ($D_0$) is illustrated in Fig. 8. The areas of individual data points are proportional to the particle counts of the corresponding observation time intervals. The overlaid black solid curve, a least squares fit applied for all cases in Table 1 is given by

$$\rho(D_0) = 0.226 D_0^{-1.004}, \tag{14}$$

where $D_0$ is in millimeters and $\rho$ is in g cm$^{-3}$. As the two examined winters were seen to have notable differences between each other in the snowfall type and average bulk density, corresponding relations were also calculated separately for the winters, and are given by

$$\rho(D_0) = 0.273 D_0^{-0.998} \quad \text{and} \tag{15}$$

$$\rho(D_0) = 0.209 D_0^{-0.969} \tag{16}$$

for BAECC events and for events of winter 2014/15, respectively. A relation by B07, given by $\rho(D) = 0.178 D_0^{-0.922}$, is plotted in Fig. 8 for comparison. As their definitions of particle diameter and bulk density close to ours, the relations are easy to compare. Especially (16) is in good agreement with B07's results. The bulk density is on average higher for snow events recorded during BAECC, which suggests more riming occurred during those events. Indication to this is that the ARM AMF2 dual-channel microwave radiometer located on the same measurement field detected the presence of liquid water more than 80% of the BAECC SNEX campaign time (Petäjä et al., 2016) and the presence of supercooled liquid layers could also be observed in the backscatter coefficient and circular depolarization ratio measurements of the co-located ARM HSRL (High



Spectral Resolution Lidar) in the majority of the BAECC cases (Goldsmith et al., 2014). In general the BAECC winter was milder than the next winter 2014–2015, and the case duration weighted average of maximum recorded temperatures was almost one degree higher for BAECC events compared to the value for winter 2014–2015 cases. The temperatures closer to 0°C could mean increased aggregation as stated in B07 and therefore decreased density values, but also different snow habits compared to more colder cases.

The mass-dimensional relation in power-law format $m = a_m D_m^b$ can be induced from the retrieved $\rho$-$D_0$ relations (14) to (16) by assuming exponential PSD and describing the bulk density as

$$\rho = \frac{\int m(D)v(D)N(D)\mathrm{d}D}{\int V(D)v(D)N(D)\mathrm{d}D} \tag{17}$$

$$= \frac{\int a_m(D)^{b_m} a_v D^{b_v} N_0 \exp(-\Lambda D)\mathrm{d}D}{\int \frac{\pi}{6}(0.1D)^3 a_v D^{b_v} N_0 \exp(-\Lambda D)\mathrm{d}D} \tag{18}$$

$$= \frac{6}{\pi} a_m \frac{\Gamma(b_m + b_v + 1)}{\Gamma(4 + b_v)} D_0^{b_m - 3} \left(\frac{0.1}{3.67}\right)^{b_m - 3}. \tag{19}$$

Taking the three velocity exponents from equations (11) to (13), the derived prefactors and exponents of mass-relation in grams are shown in Table 2, having the volume-equivalent diameter proxy in mm. The factor 0.1 in (18) derives from unit conversion, as bulk density is in $\mathrm{g\,cm^{-3}}$. The values of prefactor $a_m$ are not sensitive to the changes in the velocity exponent $b_v$, though there is a small increase in $a_m$ with increasing $b_v$. With $b_v = 0.217$ the derived mass-dimensional relations for all cases and for both studied winters separately are plotted against literature values in Fig. 9. The derived exponent $b_m$ for the studied cases is in line with literature values, close to 2, but the prefactor $a_m$ values are higher than the presented relations in Table 3. The highest value of $a_m$ is for the BAECC cases indicating conditions of riming. The high prefactor values might manifest the Finnish winter conditions, because of the vicinity of Baltic Sea, the air is more moist than e.g. in continental conditions.

### 4.3.2 $N_w$ and density

Distributions of observed $N_w$ values also exhibit dependence of $N_w$ on the bulk density, as shown in Fig. 10, i.e. $N_w$ increases with density. The modal values of $N_w$ are approximately 5000, 40,000 and 80,000 $\mathrm{mm^{-1}m^{-3}}$ for bulk density ranges $0.0\ldots0.1$, $0.1\ldots0.2$ and $>0.2\ \mathrm{g\,cm^{-3}}$, respectively, with vast majority of $N_w$ values spanning less than two orders of magnitude for a given $\rho$ range. This dependence of $N_w$ on density is somewhat unexpected. There is no obvious reason to expect that $N_w$ would depend on density. However, because $D_0$ and density are related, dependence of $N_w$ on density potentially arises from the dependence of $N_w$ on $D_0$.

A relation between $N_w$ and snow particle size is shown in Fig. 11a. A linear least squares fit is applied for $(D_0, \log(N_w))$, and the corresponding relation between $N_w$ and $D_0$ is given by

$$N_w = 2.492 \times 10^5 \times 10^{-0.620 D_0}. \tag{20}$$

Bringi and Chandrasekar (2001) show that there is a weak tendency for $N_w$ to decrease with increasing $D_0$ for rain (their Fig. 7.17), but to our knowledge, this is the first attempt to find a climatological relation between $D_0$ and $N_w$ for snow. It





should be noted, however, that the observed relation is partially caused by data filtering which removes low precipitation rate data. There is a high amount of scatter when $N_w < 1 \times 10^3 \ \mathrm{mm}^{-1}\mathrm{m}^{-3}$. The data points in this area are more contained when $D_0$ is multiplied with $\rho^{1/3}$ as shown in Fig. 11b. Making a fit to the resulting data points gives

$$N_w = 7.072 \times 10^5 \times 10^{-1.783 D_0 \rho^{1/3}} . \tag{21}$$

However, the difference in correlation coefficients for the fits in Figures 11a and b, given by -0.87 and -0.85, respectively, is minimal. The lower scatter in Fig. 11b for $N_w$ in the sub $10^3 \, \mathrm{mm}^{-1}\mathrm{m}^{-3}$ range seems to be compensated by slightly more scatter in the higher end of the distribution.

### 4.3.3  PSD shape parameter, $\mu$

In Fig. 12 the normalized frequencies of the gamma PSD shape factor $\mu$ are visualized in the three bulk density ranges. Unlike
$D_0$ and $N_w$, $\mu$ does not seem to have a clear correlation with snow bulk density, although, a weak tendency for $\mu$ to increase with density is possible. Instead, the values of $\mu$ are scattered around approximately zero, with deviation increasing with density. In the bulk density ranges 0.0 to 0.1 and 0.1 to 0.2 $\mathrm{g\,cm}^{-3}$ the kernel densities peak at -0.15 and 0.62, with standard deviations of 0.97 and 1.58, respectively. For the integration intervals with $\rho > 0.2 \, \mathrm{g\,cm}^{-3}$, the distribution of $\mu$ is more spread, with standard deviation of 2.0 and median of 0.76. The observations support the findings of B07 and Heymsfield et al. (2008),
who have found that low density particles generally have exponential or slightly super-exponential distributions. This suggests, the exponential PSD would be most appropriate for describing low density aggregated snow and less so when strong riming occurs.

## 5   Conclusions

Microphysical properties of snow in Southern Finland were documented using observations from PIP and a weighing gauge.
The data was collected during US DOE ARM funded BAECC campaign and the consecutive winter. It is shown that there is a detectable difference in measured snow properties between the two winters. Snow observed during BAECC is denser than during the next winter. The derived $m$-$D$ relations from two winters are also different, and the difference is namely in the prefactor of the power law relations.

It is found that $D_0$ and $N_w$ parameters of Gamma PSD are correlated with the bulk density. While the relation between
bulk density and $D_0$ is not surprising, since these two parameters are related, the correlation between $N_w$ and bulk density is interesting. This correlation arises from the observed connection between $N_w$ and $D_0$. It should be noted that this observed connection is partially due to data filtering that removes low precipitation rate data from the analysis. However, it indicates that for heavier precipitation aggregation is an important snow growth process. During snow growth by aggregation, $N_w$ should decrease while $D_0$ increases, as was found by (Lo and Passarelli, 1982). The shape parameter of the Gamma PSD, $\mu$, does not
seem to depend on bulk density and its average value is close to zero, which is inline with studies reported in literature.




Dependence of $v$-$D$ relation on bulk density was also studied. It was found that the prefactor of the $v$-$D$ power law depends on density. It is higher for higher densities. This result is in agreement with the conclusion made by Barthazy and Schefold (2006), that the coefficient and power terms increase with riming degree.

The presented study uses the newly developed instrument Particle Imaging Package, which is a new generation of SVI .

5   It is shown that data collected by this instrument is adequate for such studies. While the instrument only observes particle shapes projected to single 2D plane, as opposed to 2DVD or MASC, it has a larger sampling volume and its observations are less affected by wind (Newman et al., 2009). Additionally, the instrument itself is operationally more robust and requires less maintenance enabling deployment in sites with remote locations and harsh field conditions.

*Acknowledgements.*  We would like to acknowledge the Hyytiälä station and University of Helsinki personnel for the daily tasks with mea-

10   surements, especially mentioning Matti Leskinen and Janne Levula. The research of JT and DM was supported by Academy of Finland (grant 263333) and the Academy of Finland Finnish Center of Excellence program (grant 272041). AvL was funded by grant of the Vilho, Yrjö and Kalle Väisälä Foundation. The instrumentation used in this study was supported by NASA Global Precipitation Measurement Mission ground validation program and by the Office of Science U.S. Department of Energy ARM program.



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





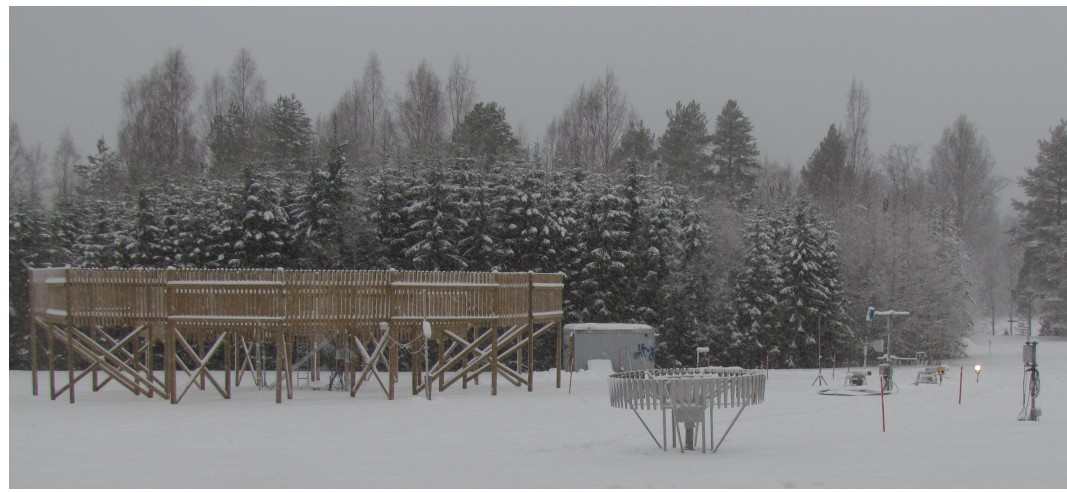

**Figure 1.** Snow precipitation instruments on the measurement field in Hyytiala. The Pluvio$^2$ 200 is inside the wind protection on a platform and the PIP lamp can be seen at right on the ground. The view of the picture is to southwest and the distance from the platform to the treeline behind is approximately 20 m.

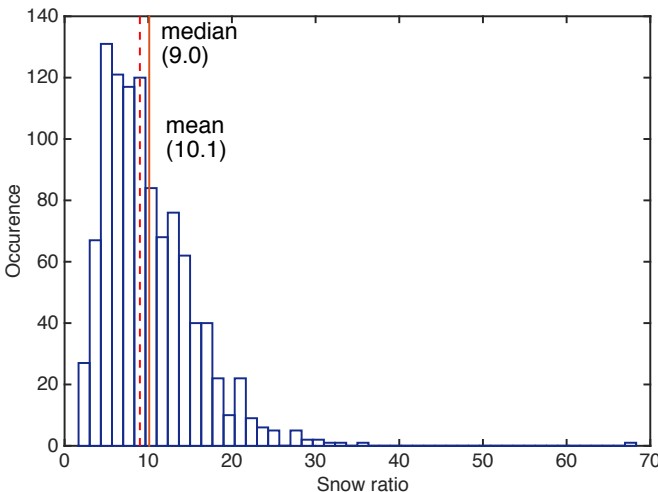

**Figure 2.** Distribution of snow ratios, ratio of snow depth change to LWE, calculated from retrieved bulk densities.





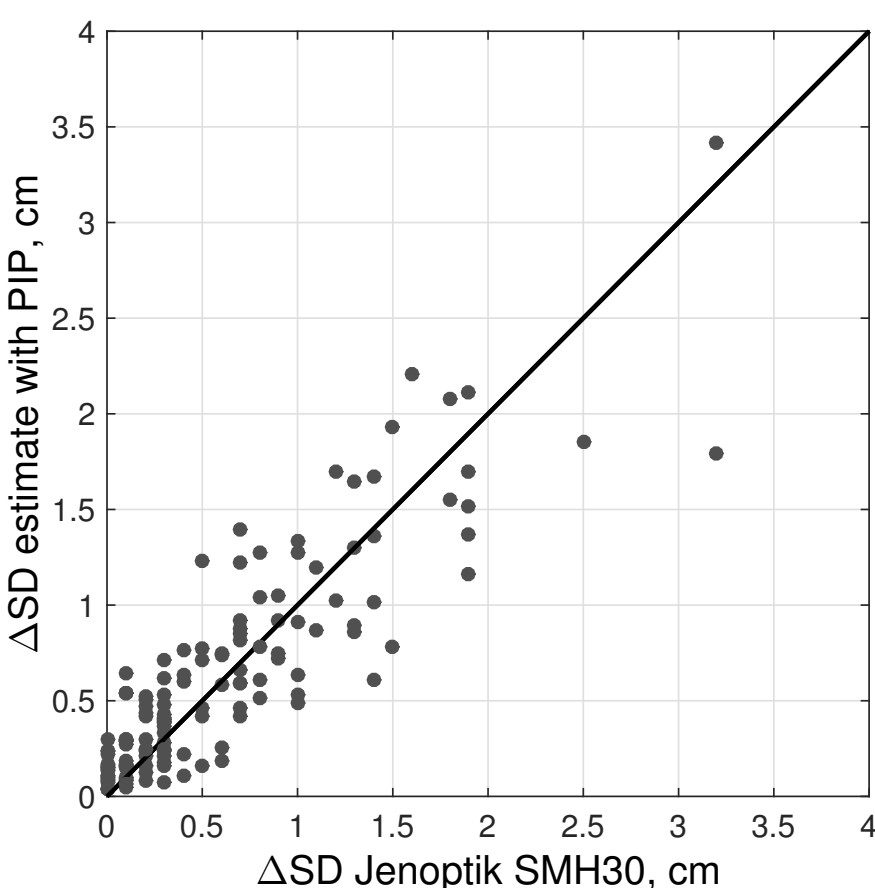

**Figure 3.** Scatterplot of the hourly change of snow depth measured with Jenoptik SMH30 and estimated from volume flux using PSD and fall velocities as measured by PIP. The data includes all the studied cases except Jan 10-11 2015.





**Figure 4.** Evolution of snowfall intensity, bulk density and particle size distribution parameters during March 18th 2015 with associated $(v, D)$ from three selected time intervals. The red lines mark the upper and lower velocity limits where for a given $D$, the KDE value is higher than half maximum.





**Figure 5.** Evolution of snowfall intensity, bulk density and particle size distribution parameters during the night between the 22nd and 23rd of January 2015 with associated $(v, D)$ from three selected time intervals. The red lines mark the upper and lower velocity limits where for a given $D$, the KDE value is higher than half maximum.



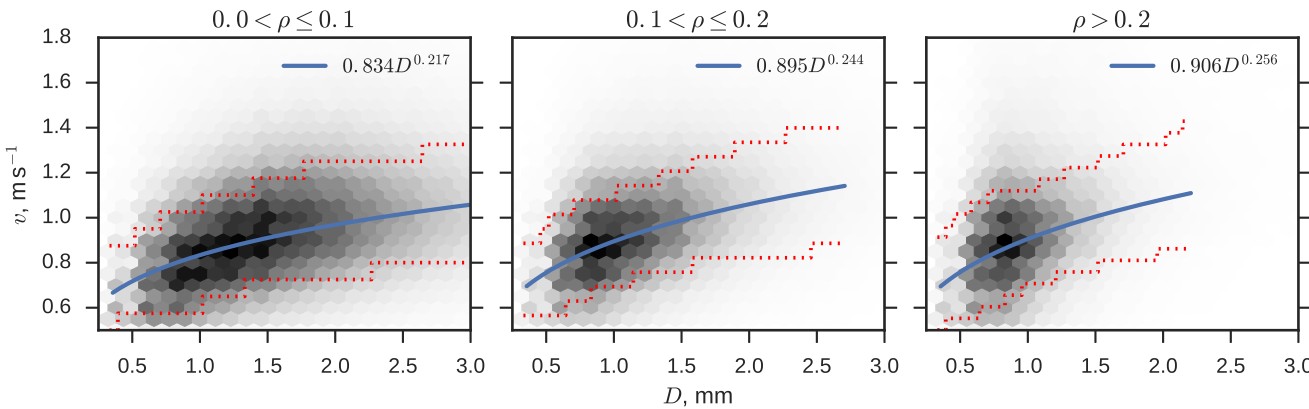

**Figure 6.** Propability densities of $(D, v)$ in three bulk density ranges. Dashed lines mark the full width at half maximum KDE in each diameter bin. Power law functions are fitted for data between those lines.





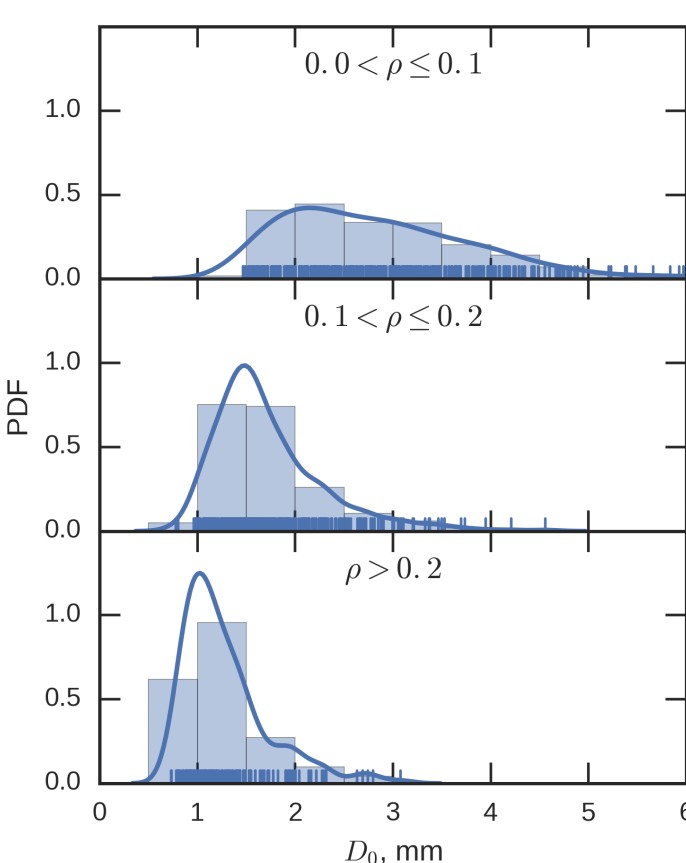

**Figure 7.** Normalized frequency (bars) and kernel density (line) of median volume diameter $D_0$ in three bulk density ranges, $[\rho] = \mathrm{g\,cm^{-3}}$.





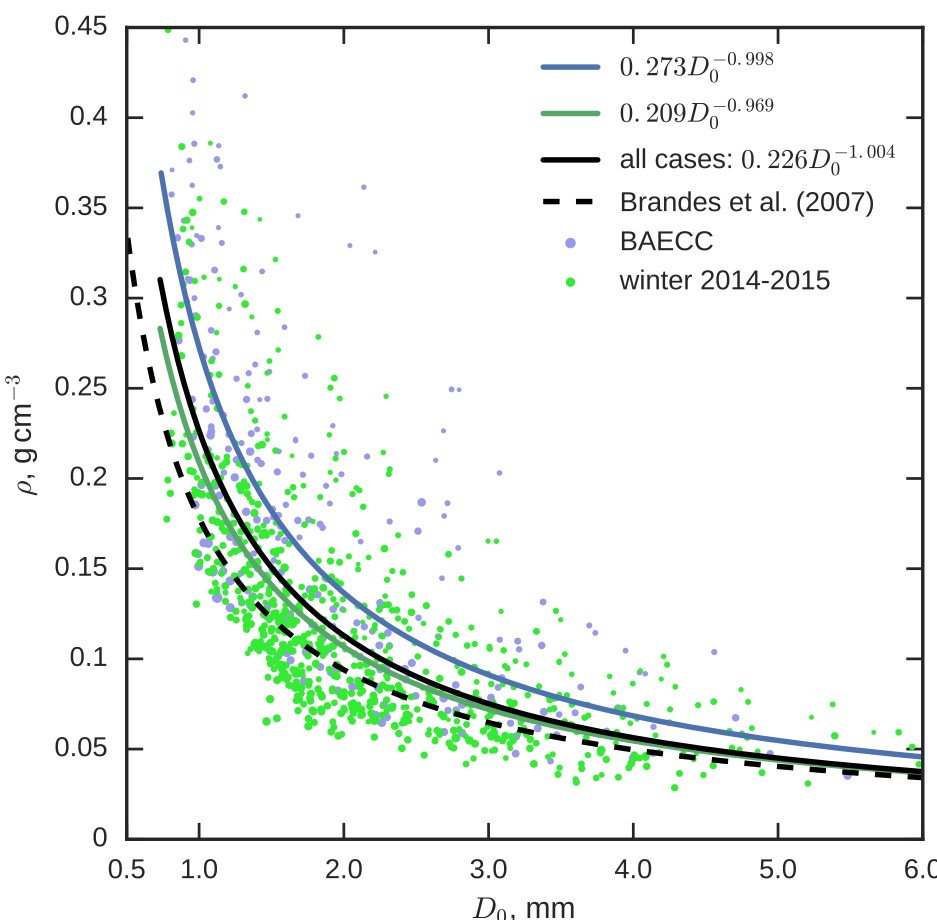

**Figure 8.** $(D_0, \rho)$ for all cases listed in table 1. Area of each dot is proportional to the number of particles in corresponding integration time interval. Power law fits are shown separately for BAECC winter cases (blue) and cases from the following winter (green).





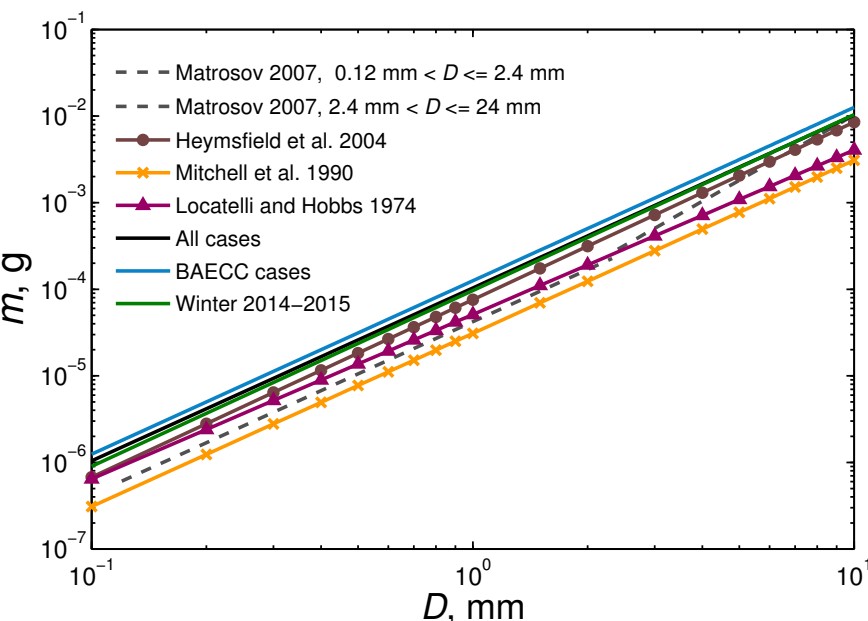

**Figure 9.** Derived $m$-$D$ relations assuming exponential PSD in comparison relations presented in literature are shown in Table 3. The conversion of maximum dimension to volume equivalent diameter is done by assuming axis ratio of 0.6.





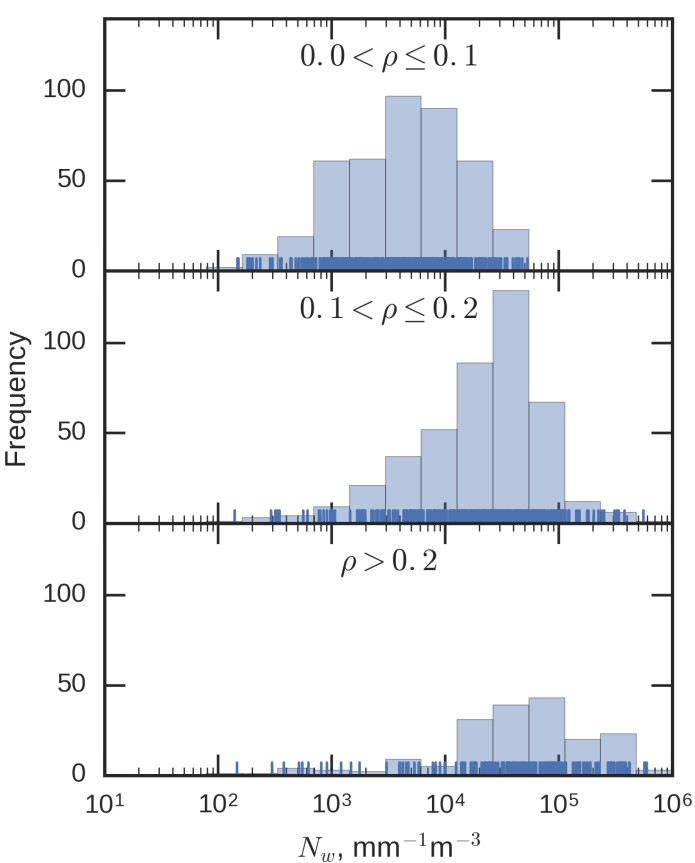

**Figure 10.** Frequency of $N_w$ in three bulk density ranges, $[\rho] = \mathrm{g\,cm}^{-3}$.





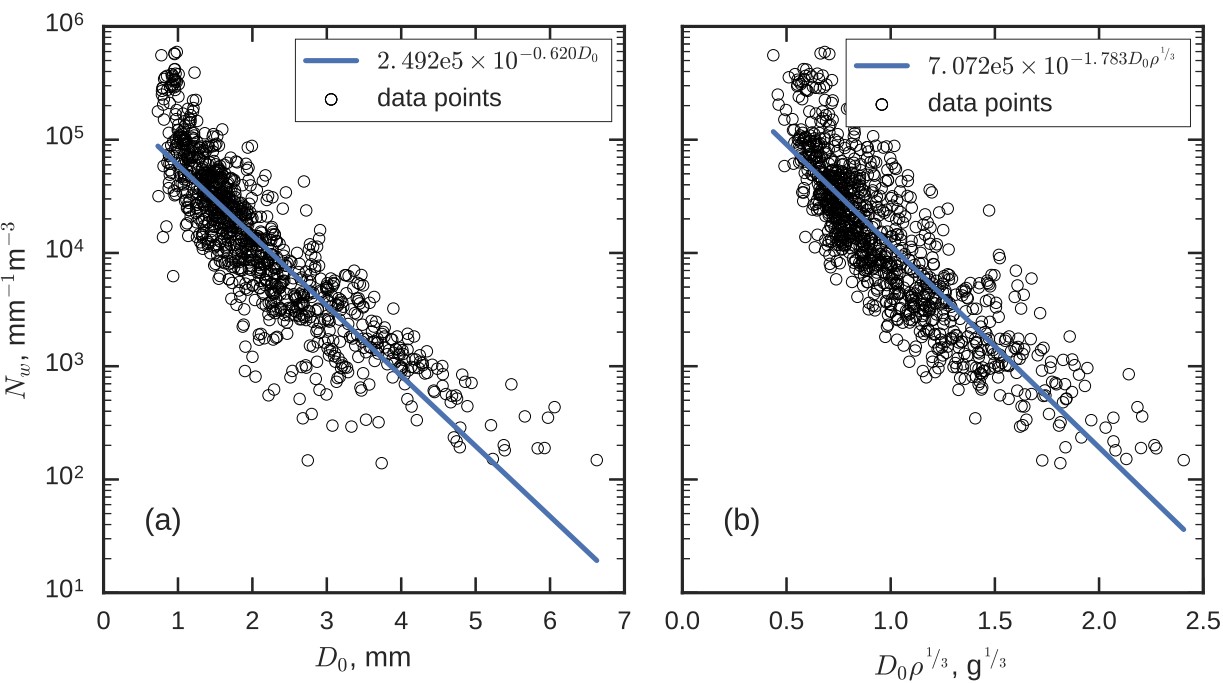

**Figure 11.** $(D_0, N_w)$ and $(D_0\rho^{1/3}, N_w)$ with fitted relations





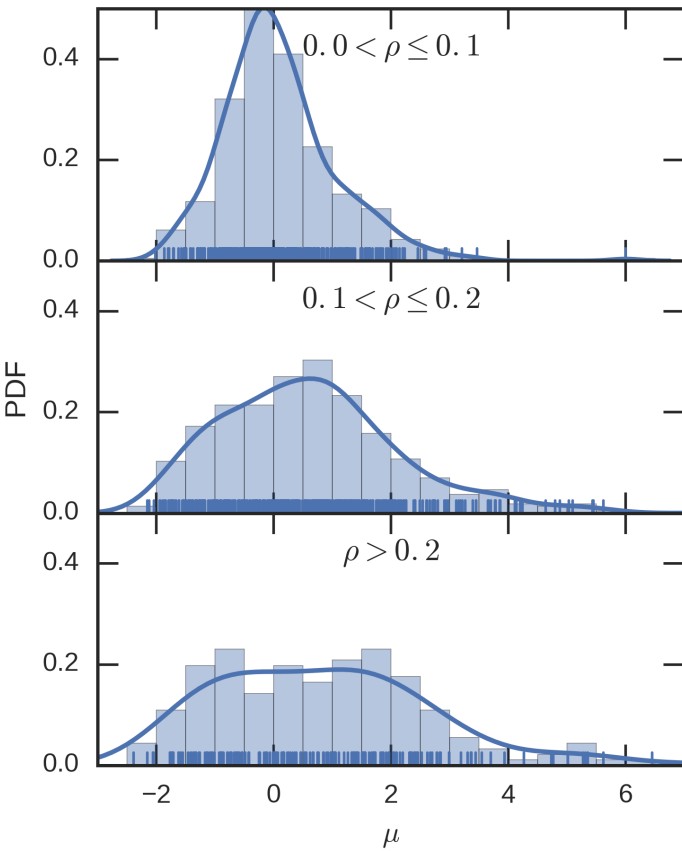

**Figure 12.** Normalized frequency (bars) and kernel density (line) of the gamma PSD shape factor $\mu$ in three bulk density ranges, $[\rho] = $ $\mathrm{g\,cm^{-3}}$.



**Table 1.** Liquid water equivalent precipitation accumulation measured with Pluvio$^2$ 200 and 400, change in snow depth and maximum and minimum temperature, maximum and minimum relative humidity, mean and maximum wind speed and mean wind direction of the studied snow events. Events before the horizontal line are recorded during the BAECC campaign.

| Event | LWE (mm) | | ΔSD | Temp (°C) | | RH (%) | | Wind (m s$^{-1}$, °) | | |
| | 200 | 400 | (cm) | min | max | min | max | mean | max | mean dir. |
| --- | --- | --- | --- | --- | --- | --- | --- | --- | --- | --- |
| 2014 Jan 31 21:00 - Feb 01 06:00 | 7.4 | 7.3 | 5.1 | -9.8 | -8.9 | 84 | 91 | 1.6 | 2.9 | 138 |
| 2014 Feb 12 04:00 - 11:00 | 1.0 | 0.9 | 1.8 | - 1 | 0 | 96 | 98 | 0.6 | 2.0 | 170 |
| 2014 Feb 15 21:00 - Feb 16 03:00 | 2.6 | 2.6 | 2.5 | -2.1 | -1 | 86 | 97 | 1.9 | 2.7 | 140 |
| 2014 Feb 21 16:00 - Feb 22 05:00 | 5.5 | 5.2 | 3.6 | -2.7 | 0 | 88 | 98 | 2.1 | 3.4 | 138 |
| 2014 Mar 18 08:00 - 19:00 | 4.4 | 4.0 | 7.3 | - 3.8 | -1.8 | 76 | 96 | 1.2 | 2.7 | 155 |
| 2014 Mar 20 16:00 - 23:00 | 6.1 | 5.9 | 4.8 | - 4.3 | -1.3 | 89 | 97 | 2.0 | 3.4 | 146 |
| 2014 Nov 06 19:00 - Nov 07 14:30 | 10.5 | – | 10.3 | -2.4 | -1.6 | 95 | 97 | 0.8 | 1.9 | 238 |
| 2014 Dec 18 14:00 - 19:00 | 2.6 | 2.2 | 3.9 | -2.3 | -0.8 | 97 | 98 | 1.0 | 1.8 | 134 |
| 2014 Dec 24 08:30 - 13:00 | 1.3 | 1.2 | 1.2 | -9.2 | -8.9 | 90 | 91 | 0.7 | 1.5 | 204 |
| 2014 Dec 30 00:30 - 14:00 | 6.3 | 5.3 | 4.9 | -10.4 | -0.6 | 91 | 98 | – | – | – |
| 2015 Jan 3 09:00 - 23:50 | 7.3 | 7.3 | 11.9 | - 3.9 | 0 | 96 | 98 | 2.6 | 5.2 | 318 |
| 2015 Jan 7 01:00 - 20:10 | 5.4 | 4.8 | 2.2 | -6.5 | -0.8 | 92 | 97 | 1.3 | 2.8 | 181 |
| 2015 Jan 8 06:00 - 13:30 | 2.6 | 2.7 | 1.6 | - 1.9 | 0 | 97 | 99 | 1.0 | 2.2 | 155 |
| 2015 Jan 9 18:00 - Jan 10 06:00 | 3.1 | 3.1 | 4.6 | -3.7 | -0.2 | 95 | 98 | 1.0 | 3.0 | 286 |
| 2015 Jan 10 22:00 - Jan 11 09:00 | 0.7 | 0.6 | 0.7 | -12.6 | -4.4 | 88 | 95 | 1.6 | 3.4 | 207 |
| 2015 Jan 12 21:00 - Jan 13 08:30 | 12.8 | 10.9 | 9.6 | -15.7 | -9.0 | 88 | 94 | 1.3 | 3.1 | 181 |
| 2015 Jan 13 22:00 - Jan 14 07:00 | –* | 2.2 | 1.9 | -8.0 | -0.3 | 94 | 98 | 0.5 | 1.9 | 134 |
| 2015 Jan 16 01:30 - 07:30 | –* | 5.8 | 5.2 | -1.3 | -0.6 | 92 | 98 | 1.9 | 3.4 | 154 |
| 2015 Jan 18 16:00 - 21:00 | 1.9 | 1.9 | 2.7 | -2.4 | -0.3 | 95 | 97 | 1.2 | 2.6 | 300 |
| 2015 Jan 22 21:00 - Jan 23 04:30 | 2.1 | 2.0 | 2.3 | -13.3 | -12.5 | 87 | 90 | – | – | – |
| 2015 Jan 23 15:00 - 23:00 | 1.4 | 1.2 | 1.4 | -10.1 | -8.8 | 91 | 93 | 0.3 | 1.0 | 205 |
| 2015 Jan 25 09:00 - 16:00 | 2.8 | 2.5 | 1.9 | -2.4 | -1.7 | 96 | 97 | 0.7 | 1.7 | 170 |
| 2015 Jan 31 12:00 - Jan 31 23:15 | 7.0 | 6.6 | 5.7 | -1.9 | -0.4 | 92 | 97 | 1.2 | 2.6 | 175 |

*Pluvio$^2$ 400 was used as data from Pluvio$^2$ 200 was unavailable



**Table 2.** The prefactors and exponents of $m = a_m D^{b_m}$ derived for exponential PSD with different values of exponent $b_v$ of velocity relation.

| Dataset | $b_m$ | $a_m(b_v = 0.217)$ | $a_m(b_v = 0.244)$ | $a_m(b_v = 0.256)$ |
|---------|-------|--------------------|--------------------|--------------------|
| All cases | 1.996 | 1.036e-4 | 1.045e-4 | 1.049e-4 |
| BAECC cases | 2.002 | 1.254e-4 | 1.264e-4 | 1.269e-4 |
| Winter 2014-2015 cases | 2.031 | 9.679e-5 | 9.757e-5 | 9.792e-5 |

**Table 3.** The prefactors and exponents of $m = a_m D^{b_m}$ of literature values for comparison plotted in Fig. 9

| Study | $b_m$ | $a_m$ |
|-------|-------|-------|
| Matrosov 2007, $0.12\,\text{mm} < D \leq 2.4\,\text{mm}$ | 2.0 | $4.2172 \times 10^{-5}$ |
| Matrosov 2007, $2.4\,\text{mm} < D \leq 24\,\text{mm}$ | 2.5 | $3.2430 \times 10^{-5}$ |
| Heymsfield et al. 2004 | 2.04 | $7.5814 \times 10^{-5}$ |
| Mitchell et al. 1990 | 2.0 | $3.0926 \times 10^{-5}$ |
| Locatelli and Hobbs 1974 | 1.9 | $5.1134 \times 10^{-5}$ |