# Peer review of "Bulk density and its connection to other microphysical properties of snow as observed in Southern Finland"

_Atmospheric Measurement Techniques, 2016_

## Referee Comment (RC1) · Anonymous Referee #1 · 26 Jun 2016

Overall it is an interesting and potentially useful paper. However, I feel that some significant revisions are needed.

General comments

1. I am concerned about adequacy of the discussion about the relation between snow depth and bulk density of falling snowflakes. The bulk density as given by (8) assumes some snowflake shape (a spheroidal shape in this case). Such snowflakes cannot be stored at the ground compactly (without air volumes between them. These internal air volumes would increase the snow depth on the ground. Snow compression on the ground might counteract this to a certain extent but the total effect is not known. Due to this I do not think the rho(t) in (9) and rho(t) in (10) are exactly the same quantities.

[Figure]

This issue needs clarification.

2. The bulk density which is sought in this study represents the whole PSD: rho=rho(Do). In many previous studies, starting probably from Magono and Naka-mura (1965), the bulk density was understood as the density of an individual snowflake defined as the ratio of the individual particle mass, which has a size D, to its volume: rho=rho(D) (for example to the spheroidal volume as this is the shape used in your study too). It causes a confusion. The Brandes et al. (2007) paper, for example, compares bulk densities from two different definitions in their table 2 and Fig. 6a. However, rho(D) is not the same as rho(Do), they are different parameters. I suggest that you clearly state different definitions of bulk densities used previously to minimize confusion for potential readers.

3. What are uncertainties of estimating bulk density and the coefficients in m-D relation? Some discussion is needed here.

4. You estimated coefficients am and bm in the m-D relation assuming the exponential distribution and just one value of bv=0.217. According to the data in Figs. 5-6, bv changes relatively widely from 0.208 to 0.256. How this variability in bv would change the derived coefficients in the m-D relations? Also what is influence of variations in the mu factor?

Specific comments.

1. Section 3.1: It appears that you model particles as oblate spheroids. Please provide some discussion to justify this model.

2. Equations (4)-(7) are obtained assuming integration from 0 to infinity. In reality there is not only truncation due to particle maximum size, but also due to the smallest considered size being 0.2 mm (not zero) as shown in the last line of page 5. Can you estimate errors due to realistic integration limits?

3. Equations (8) and (9): again why the lower integration limit is 0 (not 0.2 mm)?

4. Equations (17)-(18): put the integration limits. What would be the effect of using the non-truncated integral as given by (19) instead of smaller values which the truncation and non-zero lower limit provide.

5. How changing the assumption of the aspect ratio (currently 0.6) would change your lines in Fig. 9?

Technical corrections/comments

1. I suggest indicating in the title and introduction that you are deriving the density of FALLING snow. I tend to believe that the density of falling snow and the density of snow on the ground are different (see the general comment 1.

2. Page 3, Line 11: Indicate here which "couple of days" from Table 1 were used in this study.

3. Page 11, line 6: the subscript "m" should be at the "b" level not at "D".

4. The citation to Matrosov 2007 from Table 3 and Fig. 9 is not in the reference list.

5. Page 9, line 22: Is it "all the cases" from Table 1?

6. Add units for density in Figures 7, 10 and 12.

---

## Referee Comment (RC2) · A. Heymsfield (Referee) · 26 Jun 2016

This study uses ground-based measurements from particle probes located in Finland to look at the relationship of a population-mean ice particle density to parameters of the corresponding particle size distributions and the relationship of particle terminal velocity to diameter. The measurement techniques and their accuracies are carefully and quite fully described and, overall, the article is clearly written.

My most major concern has to do with the near-absence of a discussion of what areas would most benefit from the results presented here. It is mentioned that the properties of ice particles are an area of continuing interest for ground, airborne, and satellite remote sensing retrievals. For example, which disciplines are interested in using the "volume equivalent diameter", or "the population-mean density"? I don't see these parameters being useful for weather forecast or climate modeling because a size-dependent density is needed for these studies. To address this question, I suggest that the authors look at the citations for the Brandes article and determine what type of studies are needing these parameters.

More could have been done to estimate the density of individual particles, at least as a reality check on the ensemble-bulk densities that they do derive. Crude estimates could have been made by using particle terminal velocities and approximate cross-sectional areas (viewed from the side) and Best Number-Reynolds Number relationships that are referred to (Bohm, etc). Also, would it have been possible to derive the fractal dimension of the particles—from their cross-sectional areas and terminal velocities, such that the exponent in the mass-dimensional relationship could be estimate?

Also, if there are collocated radar observations, it would be interesting to use the PSDs and estimate densities to forward-model the radar reflectivity and compare against the meausurements.

My specific comments appear below.

1. Page 5, line 17: What was the motivation for using the equivalent area diameter?

2. Page 5, line 22: what is the approximate ratio of particle height to maximum particle width from your data set? It would be useful to show.

3. Page 7, Eq. (8). It would be good to mention here that rho is the population-mean average.

4. Page 7, line 6: "snow bulk density" to "mean snow bulk density or volume flux-weighted snow density"

5. Page 7, line 15: "used diameter" to "diameter used"

6. Page 8, line 8: From the definition of D, what would this relationship be used for?

Likewise, Page 8, Eqs. 11-13.

7. Page 10, line 12: 5...6 needs to be corrected.

8. Page 11, line 6. "induced" to "derived"

9. Page 11, Eq. (19): Eq. (19) assumes that the PSD goes from 0 to infinity. Will this assumption induce some error.

10. Page 12, line 20: consecutive winters.

11. Although you show the temperatures of the observations in Table 1, it would be good to me

---

## Referee Comment (RC3) · Anonymous Referee #3 · 28 Jun 2016

See attached file.

Please also note the supplement to this comment:
http://www.atmos-meas-tech-discuss.net/amt-2016-192/amt-2016-192-RC3-supplement.pdf

———————————————

---

## Short Comment (SC1) · 26 Jul 2016

Dear reviewer,

Thank You for Your helpful comments. After a short summer break we are currently working on giving our answers and making the necessary modifications to the manuscript. I would request clarification for Your specific comment 11. It seems like a part of the sentence is missing: "Although you show the temperatures of the observations in Table 1, it would be good to me".

Could You please rephrase this comment?

---

## Author Comment (AC1) · 26 Aug 2016

The authors appreciate the constructive criticism and helpful comments and suggestions from the Anonymous referee #1. We have addressed each of your concerns in our response below.

A marked-up version of the manuscript indicating the changes we have made is attached as a supplement to this response and all page and line numbers in our comments are in reference to that document.

**1    General comments**

*Q1.    I am concerned about adequacy of the discussion about the relation between snow depth and bulk density of falling snowflakes. The bulk density as given by (8) assumes some snowflake shape (a spheroidal shape in this case). Such snowflakes cannot be stored at the ground compactly (without air volumes between them. These internal air volumes would increase the snow depth on the ground. Snow compression on the ground might counteract this to a certain extent but the total effect is not known. Due to this I do not think the rho(t) in (9) and rho(t) in (10) are exactly the same quantities.*

**Response**. Thank you for your comment. We have added an extensive discussion on this.

**Changes**. Added a paragraph starting p.9 l.9 and some supplementary sentences in the previous and following paragraphs. Revised Eq. (10).

*Q2.    The bulk density which is sought in this study represents the whole PSD: rho=rho(Do). In many previous studies, starting probably from Magono and Nakamura (1965), the bulk density was understood as the density of an individual snowflake defined as the ratio of the individual particle mass, which has a size D, to its volume: rho=rho(D) (for example to the spheroidal volume as this is the shape used in your study too). It causes a confusion. The Brandes et al. (2007) paper, for example, compares bulk densities from two different definitions in their table 2 and Fig. 6a. However, rho(D) is not the same as rho(Do), they are different parameters. I suggest that you clearly state different definitions of bulk densities used previously to minimize confusion for potential readers.*

**Response**. It is a very good point. We have added discussion about the different definitions, and the density related terminology has been partly changed and more

accurately described. To avoid confusion with the term "bulk density" and its multiple definitions, we chose to use the term "ensemble mean density" instead. As of notation, $\bar{\rho}$ is now used to denote ensemble mean density instead of $\rho$.

**Changes**. Added discussion starting p.2 l.26.

*Q3. What are uncertainties of estimating bulk density and the coefficients in m-D relation? Some discussion is needed here.*

**Response**. We have added more discussion and performed a simulation study to quantify some of the uncertainties.

**Changes**. Uncertainties due to PSD truncation are discussed in the new Chapter 3.4 (p.9, l.35) and from p.14 l.17 onwards.

*Q4. You estimated coefficients $a_m$ and $b_m$ in the m-D relation assuming the exponential distribution and just one value of bv=0.217. According to the data in Figs. 5-6, bv changes relatively widely from 0.208 to 0.256. How this variability in bv would change the derived coefficients in the m-D relations? Also what is influence of variations in the mu factor?*

**Response**. We have computed the $m\text{-}D$ relations to the different exponent values $b_v$ of $v$ - $D$ relations defined in Equations 11-13 for different bulk density ranges and the results are stated in Table 2. It can be seen that the prefactor values $a_m$ are not sensitive to changes of $b_v$ (this is stated on p.14 l.27 of the revised manuscript), $a_m$ values change less than 1% as the values of $b_v$ deviate. In respect to influence of $\mu$, by deviating $\mu$ from values of 0 to 3, the $a_m$ is changing nearly 48%, increasing as $\mu$ is increasing. The influence of $\mu$ is now described starting p.14, l.28.

**2 Specific comments**

*Q1. Section 3.1: It appears that you model particles as oblate spheroids. Please provide some discussion to justify this model.*

**Response**. We have added a new figure, the new Fig. 2, and more discussion, especially in the Section 3.1.

**Changes**. Discussion starting p.6, l.8.

*Q2. Equations (4)-(7) are obtained assuming integration from 0 to infinity. In reality there is not only truncation due to particle maximum size, but also due to the smallest considered size being 0.2 mm (not zero) as shown in the last line of page 5. Can you estimate errors due to realistic integration limits?*

**Response**. We have performed a simulation study to quantify this effect. The results of this study are summarized in Fig. 6. As you can see, we are overestimating density. For $D_0$ larger 1 mm, most of our observations, this overestimation is below 5%.

**Changes**. New Chapter 3.4 (starting p.9)

*Q3. Equations (8) and (9): again why the lower integration limit is 0 (not 0.2 mm)?*

**Response**. It is now changed to $D_{min}$.

*Q4. Equations (17)-(18): put the integration limits. What would be the effect of using the non-truncated integral as given by (19) instead of smaller values which the truncation and non-zero lower limit provide.*

**Response**. it is discussed in more detail now, see answer to Q2.

*Q5. How changing the assumption of the aspect ratio (currently 0.6) would change your lines in Fig. 9?*

**Response**. The change in the aspect ratio would have a noticeable effect on the lines in Fig. 9 (new Fig. 10), at least if the m-D relation would be retrieved from a single event. The impact of the aspect ratio assumption on the retrieved density is now discussed in the Section 3.1. (starting p.6, l.15) However, since we are using measurements from several events, and comparison to snow depth measurements indicates that there are no large systematic biases, we are believe that presented average m-D relations are valid.

**3 Technical comments**

*Q1. I suggest indicating in the title and introduction that you are deriving the density of FALLING snow. I tend to believe that the density of falling snow and the density of snow on the ground are different (see the general comment 1.*

**Response**. Changed "snow" to "falling snow" in the title.

*Q2. Page 3, Line 11: Indicate here which "couple of days" from Table 1 were used in this study.*

**Response**. The events that "extended over a couple of days" referred to events that extended past midnight (UTC). The superfluous clause "where a number of events extended over a couple of days" was removed to avoid confusion.

*Q3. Page 11, line 6: the subscript "m" should be at the "b" level not at "D".*

**Response**. Corrected.

*Q4. The citation to Matrosov 2007 from Table 3 and Fig. 9 is not in the reference list.*

**Response**. Corrected.

*Q5. Page 9, line 22: Is it "all the cases" from Table 1?*

**Response**. Yes. It is now written "all the cases in Table 1".

*Q6. Add units for density in Figures 7, 10 and 12.*

**Response**. The units are given in the captions to save space in the figures.

**Supplement:**

**Ensemble mean density and its connection to other microphysical properties of falling snow as observed in Southern Finland**

[revised manuscript text omitted]

By combining optical disdrometer observations with other measurements, e.g. by radar or precipitation gauge, physical properties such as mean snow density can be derived. Huang et al. (2010) have used a C-band weather radar observations of equivalent reflectivity factor, $Z_e$, in combination with a 2DVD to derive snow density-dimensional relation and to infer more consistent $Z_e$–snowfall rate, SR, relations. Another method for snow density retrieval is based on solving aerodynamic equations to derive particle mass from observed fall velocity and particle effective projected area as proposed by Böhm (1989) and applied by Hanesch (1999) and more recently by Szyrmer and Zawadzki (2010) and Huang et al. (2015). Brandes et al. (2007), hereafter referred to as B07, used a combination of a weighing gauge and a 2DVD to derive mean bulk density-median volume diameter relations and to document relations between PSD parameters for Colorado winter storms. Their approach is similar to the one used by Heymsfield et al. (2004) who have combined aircraft PSD and ice water content observations to derive  mean snow density and average mass-dimensional relations for ice particles. Albeit using slightly different definitions, both B07 and Heymsfield et al. (2004) derive effective ice densities for ensembles of ice particles, but there is a difference in terminology.  Heymsfield et al. (2004) and many others have used the term (particle) bulk density to refer to the ~~derived densities as bulk densities whereas Heymsfield et al. (2004) talk about effective densities of particle populations. Another method for snow density retrieval is based on solving aerodynamic equations to derive particle mass from observed fall velocity and particle effective projected area as proposed by Böhm (1989) and applied by Hanesch (1999) and more recently by Szyrmer and Zawadzki (2010) and Huang et al. (2015)~~ density of individual ice or snow particles defined as the ratio of mass of a particle having a size $D$ to its assumed volume: $\rho = \rho(D)$. In most of such cases, the word "bulk" is used to emphasize the inclusion of hollows within particles. The term "(mean) bulk density" is sometimes used also when referring to the mean density of an ensemble of particles representing the whole PSD, i.e. $\bar{\rho} = \bar{\rho}(D_0)$ (e.g., B07), whereas Heymsfield et al. (2004) used the term population-mean effective density. In this study we

derive the volume flux weighted snow density, similar to e.g. B07, and refer to it as ensemble mean density, $\bar{\rho}$, to avoid possible confusion.

This paper documents connection between  ensemble mean density and other microphysical properties of snow as observed in Southern Finland. Using the estimated $\bar{\rho}$, average mass-dimensional relations characteristic to studied snowfall events are defined. In order to derive  ensemble mean density, a method proposed by B07 was used. However, instead of a 2DVD, a new generation of the SVI is employed. It is shown that, despite simpler construction compared to the 2DVD, this instrument's data is suitable for such studies.

Even though this study is based on retrieval of ensemble mean snow density and not mass-dimensional relations directly, which could be more easily applied to radar retrievals and numerical weather prediction (NWP). There are a number of applications of such relations. Aikins et al. (2016) used $\bar{\rho}(D_0)$ to convert particle size distribution observations to precipitation rate. Tong and Xue (2008); Dolan and Rutledge (2009); Matrosov et al. (2009); Huang et al. (2010); Zhang et al. (2011) used mean snow density-median volume diameter relations for characterizing winter precipitation microphysics by radar. Kneifel et al. (2015) showed a connection between mean snow density and multi-frequency radar observations. Thompson et al. (2008) used the density relation by B07, and Iguchi et al. (2012) applied a similar density retrieval method to improve parametrization of snow microphyics in NWP models, for example.

**2    Measurements**

**2.1    Measurement setup**

[revised manuscript text omitted]

**2.4 Snow depth sensor**

The laser snow depth sensor, Jenoptik SHM30, is located on the measurement field, next to Pluvio$^2$ 400. It is an  optical sensor, which measures the snow depth by comparing signal phase information of the modulated visible laser light. It is a point measurement, hence the piling of wind driven snow or random branches and leaves drifting on the snow pack can cause misreadings. To reduce this we have sheltered the measurement spot with a small wind fence and the instrument structure excluding the measurement pole is buried under the ground to prevent the piling of snow. The data is recorded every minute.

**3 Retrievals of  ensemble mean density, velocity-dimensional relations and PSD**

Observations from the PIP and one of the weighing gauges are combined to retrieve  snow ensemble mean density. Typically the gauge located inside of the DFIR, the Pluvio$^2$ 200, is used for this retrieval. On a couple of days this gauge was not operational and data from the Pluvio$^2$ 400  located outside of the DFIR was used instead. These dates are marked in Table 1 with asterisks in the LWE precipitation rate column. As seen in the Table 1 the differences in accumulated LWE recorded by the two Pluvio$^2$s are small, largest being 15 %. Pluvio$^2$ 200 inside the DFIR is typically measuring higher accumulations, which is expected because of the  better wind protection. However, the observations do not show a clear indication that  the observed precipitaiton accumulation difference depends on the wind speed. However, the difference seems to increase in respect to certain wind directions. There are two openings from the measurement field, one to a road crossing (approx. 130°) and the other to small field (approx. 180°). If the wind is blowing from these directions the difference between the two gauges seem to increase.

The retrieval procedure is described below and is similar to the one presented by B07, but with notable modifications. Prior to retrieval of  $\bar{\rho}$, PSD and velocity-dimensional relations are estimated. It was found, however, that the density retrieval is highly sensitive to the integration time. To minimize this, a variable integraiton time determined by the precipitation accumulation is used. The same integration time was applied to compute PSD parameters and $v$-$D$ relations.

**3.1 Particle size distribution**

The PSDs are calculated from the PIP records of particles that fell through the observation volume. The observed distributions are defined with respect to equivalent area diameter $D_{\mathrm{PIP}}$, which is different from the apparent diameter of the 2DVD

5 and maximum particle dimensions used in other studies (e.g., Heymsfield et al., 2004). Wood et al. (2013) studied differences between diameter definitions and found that the diameter recorded by SVI is approximately 0.82 of maximum particle dimension. We performed a similar study by examining mean dimensions of  rotated ellipsoids on a single projection, as shown in Fig. 2. The ellipsoids were defined by a long dimension $a$ and a short dimension $b$ lying nominally in the horizontal plane along the $x$ and  $y$-axes, respectively, and

10 a short vertical dimension $c$ lying nominally along the $z$-axis. The particle orientation was defined by Gaussian distribution of canting angles with a standard deviation of 9° (Matrosov et al., 2005a) and a uniform distribution of azimuth angles. The equivalent area diameters $D_{\mathrm{PIP}}$ of simulated particles were estimated as their projected areas onto the $x$-$z$ plane and the resulting values were averaged over all orientations. The ratios of mean $D_{\mathrm{PIP}}$ to the particle volume equivalent diameter, i.e. the diameter for which the particle volume $V(D) = \frac{\pi}{6}D^3$

15 , for a number of combinations of vertical and horizontal aspect ratios are shown in Fig. 2. Assuming spheroids (Matrosov, 2007) and taking the typical vertical aspect ratio $c/a = 0.6$ (Korolev and Isaac, 2003; Matrosov et al., 2005b) we found that $D_{\mathrm{PIP}}$ is roughly equal to 0.92 of a volume equivalent diameter. As can be seen, the conversion factor varies between 0.8 and 1. For ice particles with axis ratios smaller than 0.4, i.e. pristine ice crystals, this factor could approach 1.4. From this analysis we can conclude that the largest expected error is associated with observations of ice crystals. Dimensions

20 of snowflake aggregates and graupel like particles are expected to be captured with a smaller error. In this study the same conversion factor of  0.92 is used for all the cases. As can be seen in Fig. 3 the median area and aspect ratios of the particles are 0.65 and  0.72, respectively. These observations also support our choice of a mean particle shape and the corresponding

25 diameter transformation. Therefore, the results presented in the rest of the manuscript are using this volume equivalent diameter proxy.

Prior to calculations of PSD parameters, recorded PSD data is filtered to remove spurious observations of large particles. Following the procedure described in  ?, records of large particles were ignored if there was a gap of more than three consecutive PSD diameter bins. The bin size was set to 0.25 mm during the BAECC experiment and it was reduced to 0.2 mm for the winter 2014/2015. The PIP resolution is 0.1 mm and the minimum detectable particle diameter is approximately 0.3 mm (Newman et al., 2009). The smallest diameter bin used in calculations is 0.25 mm to 0.5 mm during BAECC and 0.2 mm to 0.4 mm in the following winter.

5 The PSD parameters were calculated using method of moments and assuming that PSD follows gamma functional form, see for example Ulbrich and Atlas (1998) and citations therein. The normalized gamma distribution $N(D)$ in $\mathrm{mm}^{-1}\mathrm{m}^{-3}$ was adopted following Testud et al. (2001); Bringi and Chandrasekar (2001); Illingworth and Blackman (2002):

$$N(D) = N_w f(\mu) \left(\frac{D}{D_0}\right)^\mu \exp(-\Lambda D) \tag{1}$$

$$f(\mu) = \frac{6}{3.67^4} \frac{(3.67+\mu)^{\mu+4}}{\Gamma(\mu+4)} \tag{2}$$

$$\Lambda = \frac{3.67+\mu}{D_0}, \tag{3}$$

with $N_w$ in $\mathrm{mm^{-1}m^{-3}}$ being the intercept parameter, $D_0$ the median volume diameter in mm, $\Lambda$ the slope parameter in $\mathrm{mm^{-1}}$ and $\mu$ the shape parameter. Using the second, fourth and sixth moments for the non-truncated gamma PSD, $M_2$, $M_4$, and $M_6$, the PSD parameters were estimated as follows:

$$\eta = \frac{M_4^2}{M_6 M_2} \tag{4}$$

$$\mu = \frac{7 - 11\eta - \sqrt{\eta^2 + 14\eta + 1}}{2(\eta-1)} \tag{5}$$

$$\Lambda = \sqrt{\frac{M_2 \Gamma(\mu+5)}{M_4 \Gamma(\mu+3)}} \tag{6}$$

$$D_0 = \frac{3.67+\mu}{\Lambda} \tag{7}$$

**3.2  Ensemble mean density retrieval**

The integration time, $\tau(t)$, of the  ensemble mean density retrieval is driven by precipitation measurements of the Pluvio[2]. The step of the non-real-time accumulation output is 0.05 mm, causing the output interval to be in the order of several minutes even at moderate snow rates. With a short fixed integration time in time scales of minutes or tens of minutes, the produced  ensemble mean density estimation would hence be the more unstable, the lower the precipitation rate. Therefore, variable length time intervals driven by the gauge output are used with a selected threshold value of 0.1 mm. This corresponds to a $\tau(t)$ of 6 minutes for a LWE precipitation intensity of $1\ \mathrm{mm\,h^{-1}}$. Effectively, the temporal resolution of the ensemble mean density retrieval is increased with increasing precipitation intensity, and in the analysis of the snowfall events in Table 1, the median $\tau(t)$ was 5 minutes.

As the integration time  $\tau(t)$ is effectively driven by precipitation intensity, there is less variation in number of particles between intervals, compared to a fixed time interval approach. With the selected accumulation threshold there are typically between $10^3$ and $10^4$ particles within a given integration time interval. On the other hand, with low precipitation intensities, $\tau(t)$ increases up to one hour and retrieved  $\bar{\rho}$ becomes less representative for the time interval in question. With LWE precipitation rates lower than $0.2\ \mathrm{mm\,h^{-1}}$, the resolution of Pluvio[2] LWE measurements is insufficient and calculations of  $\bar{\rho}$ become overly sensitive to recorded number concentrations. Correspondingly, similar unwanted sensitivity to LWE precipitation accumulation occurs when the number of particles observed by PIP within $\tau(t)$ is less than 800. Therefore, time intervals with precipitation rates or particle counts lower than these thresholds are excluded from our analysis.

Given a population of solid precipitation particles with volume equivalent diameters $D$ over the integration time $\tau(t)$, the liquid equivalent precipitation accumulation in mm is approximately

$$G(t) \approx \frac{\pi}{6} \times 10^{-6} \frac{\bar{\rho}}{\rho_w} \int_t^{t+\tau(t)} \int_{D_{min}}^{D_{max}} D^3 v(D,t) N(D,t) \, \mathrm{d}D \mathrm{d}t, \tag{8}$$

where $\bar{\rho}$ is the volume flux weighted population mean snow density in $\mathrm{g\,cm^{-3}}$, $\rho_w = 1\ \mathrm{g\,cm^{-3}}$ is the density of liquid water, $N(D,t)$ is mean particle number concentration over the integration time in $\mathrm{mm^{-1}m^{-3}}$, $v(D,t)$ is particle velocity relation in $\mathrm{m\,s^{-1}}$ and $[D_{min}, D_{max}]$ is the size range of snowflake observations from a disdrometer. From (8) we can estimate volume flux weighted snow density for each observation time interval as

$$\bar{\rho}(t) \approx \frac{6}{\pi} \times 10^6 \rho_w \frac{G(t)}{\int_t^{t+\tau(t)} \int_{D_{min}}^{D_{max}} D^3 v(D,t) N(D,t) \, \mathrm{d}D \mathrm{d}t}, \tag{9}$$

using liquid equivalent precipitation accumulation $G(t)$ as measured by the Pluvio$^2$ gauge, and retrieving volume flux with fitted $v(D,t)$, and averaged $N(D,t)$ as measured by the PIP. It should be noted, that unlike in the retrieval of PSD parameters, where gamma PSD was assumed, $\bar{\rho}$ was retrieved without making any assumptions on the shape of the PSD distribution, and instead, measured PSD are used in the calculations.

**3.3 Comparison of derived mean density to snow depth observations**

The definition of ensemble mean density here is the same as for mean bulk density in B07. They determine the densities for 5-minute precipitation volumes derived with a 2DVD disdrometer observations together with precipitation mass measured by a weighing gauge. B07 defined the volume of a single particle by summing coin-shaped sub-volumes together estimated separately for both orthogonal projections and taking geometrical mean. As the diameter used in our study is the estimated volume-equivalent diameter, our results are comparable to B07. In Heymsfield et al. (2004), the volume of a single particle is defined as a function of circumscribing maximum diameter, and the population mean effective density is determined from ice water content (IWC). The estimated ensemble mean snow density is volume-weighted and expected to have lower values than the velocity-weighted snow density. The difference is not generally prominent especially with low-density aggregates, whose velocity-dimensional dependence is weak.

It should be noted that the derived density is inversely proportional to the snow ratio, $R_s$, assuming that issues related to packing of snowflakes on the ground can be ignored. The snow ratio (Power et al., 1964; Ware et al., 2006) is used by operational weather services to estimate change in snow depth from LWE observations and can be defined as follows:

$$R_s(t) = \frac{\rho_w}{\bar{\rho}(t)} \frac{1}{P \cdot C} \tag{10}$$

where $\bar{\rho}(t)$ is the volume flux weighted snow density derived as shown in (9), $P$ is the packing efficiency of snowflakes and $C$ is the snow compression. Assuming that the packing and compression terms, or their product

are close to unity, the derived density can be  tested against the commonly used assumption that 1 mm of LWE accumulation corresponds to 1 cm change in snow depth. In Fig. 4 the combined distribution of estimated snow ratios on temporal scales defined by the gauge accumulation for all the 23 events analyzed in this study is shown. It can be seen that the mean and median values, equal to 10 and 9 respectively, are very close to the commonly assumed value.

This analysis assumes that packing efficiency of snowflakes is 100 % and compression of snow on the ground can be ignored, or snow compression counteracts reduction in snow density due to packing. The packing efficiency of snowflakes on the ground is not known. Random packing of the same size spheres has density of 64 % and dense packing of such spheres uses 74 % of the volume, corresponding to $P = 0.64$ and $0.74$ respectively. Packing efficiency of equal spheroids depends on axis ratios and exceeds this of spheres and could exceed 77 % (Donev et al., 2004) . It is not unreasonable to expect that irregular shaped particles of variable sizes, such as snowflakes, would pack more efficiently than equal spheroids. At least, packing efficiency in excess of 90 % can be expected for spheres of several radii (de Laat et al., 2014) . The packing efficiency of 70 % would mean that density of freshly fallen snow would be 30 % lower than this of falling snowflakes. The packing efficiency of 80 % would correspond to 20 % bias in estimated snowflake density from snow depth measurements, or in 25 % underestimation of the snow depth change by using $\bar{\rho}(t)$. We don't know the exact value of the snow packing, but could expect that in the worst case scenario it is about 70 % and probably closer to 80 % or even higher. It should also be noted that the snow compression would counteract this, but we are considering only freshly fallen snow and expect that the compression factor $C$ is very close to unity.

One of the major uncertainties in the density retrieval is the assumption about particle volume. In this study we have assumed that snowflakes are spheroids with axis ratios of 0.6. Given this assumption, a conversion factor relating volume equivalent and observed disc equivalent diameters was defined. Fig. 2 shows that for a reasonable range of ellipsoid axis ratios this conversion factor can range between 0.8 and 1. This range of values implies that the uncertainty in the density estimation can range from an overestimation by as much as 50 % to an underestimation by about 20 %. This range of uncertainty is much larger than what is expected from a comparison of the retrieved volume-flux weighted density and snow depth measurements, as was discussed above. Therefore, by comparing the PIP derived and the directly measured snow depths, the validity of the derived  values of $\bar{\rho}$, and assumption of particle shape, can be checked. In Fig. 5 hourly change in the snow depth measured by the Jenoptik SHD30 is compared to the PIP derived snow depth. It can be seen that the agreement is good, with RMSE of 0.30 cm, linear correlation coefficient of 0.88 and normalized bias as low as $-0.06$. This comparison also gives confidence about the validity of the derived  ensemble mean densities.

**3.4 Effect of PSD truncation on derived ensemble mean snow density**

The observed PSD are truncated on left and right sides (Ulbrich and Atlas, 1998) . They are truncated on the right side because of the instrument finite sampling volume and because natural sizes of hydrometeors do not extend to infinity. The truncation on the left, on the small-diameter side, is due to instrumental limitations and possible wind effects (Moisseev and Chandrasekar, 2007) .

Ulbrich and Atlas (1998) have presented a comprehensive analysis on how the right-side truncation affects the derived Gamma PSD parameters. A similar study on the effects of the left-side truncation and other instrumental effects was presented by Moisseev and Chandrasekar (2007). Here we apply the method presented by Moisseev and Chandrasekar (2007) to estimate impact of PSD truncation on the derived mean snow density.

To investigate the impact of the PSD truncation on retrieval of mean snow density a simulation study was performed. To initiate the simulation the PSD parameters $N_w$, $D_0$ and $\mu$, together with parameters of $m$-$D$ and $v$-$D$ are used. During the study it was found that the density estimation error is most sensitive to $D_0$ and $\mu$ and virtually independent of the other input parameters. Therefore, results presented here assume that $N_w$ is constant and equal to $10^4$ mm$^{-1}$m$^{-3}$, only one $m$-$D$ relation representative of all BAECC cases, as presented in Section 4.3.1, is selected and $v$-$D$ representative of the snowfall with mean density ranging between 100 and 200 g cm$^{-3}$ is utilized. The $D_0$ values were varied between 0.5 and 4 and $\mu$ values between $-0.9$ and 3.

At the first stage of simulation a number of observed particles were computed assuming that it follows a Poisson distribution and the expected number of particles is determined by PIP sampling volume and the integration time that is determined by the precipitation accumulation. Given this number of particles, their diameters were found by sampling a Gamma probability density function, parameters of which are determined by the input PSD. To simulate the left-side truncation all particles with diameters smaller or equal to 0.25 mm, the PIP sensitivity threshold, were rejected. The right-side truncation was achieved by rejecting particles with sizes exceeding $3D_0$. For each $D_0$ and $\mu$ pair 50 simulated PSD were computed. Given the simulated truncated PSD the density is estimated in the same way as was presented above. This estimated density is compared to the one that is directly derived from the simulation input parameters and the results of their comparison is shown in Fig. 6. As one can see, the derived ensemble mean snow density is biased. The bias is largest for small $D_0$, which is explained by the left-side PSD truncation. For $D_0$ larger than 1 mm the bias decreases and approaches 2 %. Given that the error associated with PSD truncation is rather small, at least for $D_0$ larger than 1 mm and most of our observation fall within this range, in this study the truncation error is not corrected.

**3.5 Velocity-dimensional analysis**

For the retrieval of volume flux weighted snow density, velocity-dimensional relations of falling snow need to be estimated. For each integration time interval, $v(D) = a_v D^{b_v}$ is fitted for velocity-diameter data from the PIP. The $v(D)$ power law fits to unfiltered data tend to be strongly biased by outliers. To address this problem, Gaussian kernel density estimation (KDE, Silverman, 1986) is used to find the most probable velocity for each diameter bin, and only observations with velocities within half width at half maximum from the bin peak KDE value are included in calculating the fit. Using the linear least squares method, a fit is performed for the data points in log-log scale to derive a power law relation.  It should be noted, that using linear regression in log-log space does not optimally minimize residuals in linear space, but the method is used here as it does not overly emphasize the large end of the size spectrum. The retrieved velocity fits are shown for selected integration time intervals of the 18 March 2014 and the 22-23 January 2015 cases in the bottom of Figures 7 and 8, respectively.

It should be noted that the power law model, albeit widely used, may not necessary represent correctly velocities of ice particles over the complete range of diameters (Mitchell and Heymsfield, 2005). In many cases the fit can also be uncertain
10 either because of narrow PSD or in presence of multiple particle types.

**4   Results**

**4.1   Case studies**

**4.1.1   18 March 2014**

During the March 18, Finland was covered in a continental polar air mass. In the morning, a warm occluded front associated
15 with a weak low pressure center approached southern Finland from southwest bringing light snowfall. In the afternoon Hyytiälä was in the warm sector of the frontal system, and the relative humidity dropped halting the snowfall around 12 UTC. Later in the evening there was a one-hour snow shower from a squall line associated with a cold front passing over southern Finland.

Time series of LWE snow rate,  ensemble mean density and PSD parameters for the March 18 case are shown in Fig. 7. The bottom panels show measured fall velocities for selected integration time intervals representing observations with
20 different  ensemble mean densities. Between the red dotted lines is the region where KDE is higher than half maximum for a given particle size. The fits are applied for data points between these lines. There is considerable scatter in particle fall velocity throughout the case and a bimodal PSD is present momentarily in the morning as can be seen in fall velocity panel Fig. 7a.

During the snow shower in the evening, liquid equivalent precipitation rates were recorded on average roughly three times
25 more intense than earlier during the day , allowing retrievals of  $\bar{\rho}$ and PSD parameters at high time resolutions. Strong short time scale variations of  $\bar{\rho}$ and PSD parameters are recorded during this shower. The lowest  ensemble mean density value of the case, 0.035 g cm$^{-3}$, is retrieved for time interval from 16:35 to 16:39, with concurrent $D_0$ value of 5.5 mm and $N_w$ of roughly 700 mm$^{-1}$m$^{-3}$. The corresponding fall velocity distribution visualized in panel 7b is characterized by low values of velocity fit coefficients $a_v$ and $b_v$. Within the following 20 minutes, $D_0$ decreases down to
30 roughly 2 mm, $N_w$ increases to $2 \times 10^4$ mm$^{-1}$m$^{-3}$, and retrieved values of  $\bar{\rho}$ peak at over 0.2 g cm$^{-3}$ between 16:54 and 16:58, and again from 17:05 to 17:08. Corresponding fall velocity distribution between 16:54 and 16:56, shown in panel 7c, is characterized by substantially higher values of $a_v$ and $b_v$, than 20 minutes earlier.

**4.1.2   22-23 January 2015**

During 22 January 2015, similarly to the 18 March 2014 event, a warm occluded front associated with a weak low moved northwards over the Gulf of Finland. However, due to a blocking high over north-western Russia, the low and the associated
5 front were sustained over southern Finland for the whole day of January 23rd causing weak continuous precipitation in the area.

Time series of LWE snow rate,  $\bar{\rho}$ and PSD parameters for the 22-23 January 2015 case, with velocity-diameter fits from selected time intervals are shown in Fig. 8. The case is characterized by continuous snowfall at LWE precipitation rates lower than 1 mm h$^{-1}$ throughout the case. The velocity distribution for a given time interval has substantially less scatter compared to the 18 March 2014 case. The evolution of  $\bar{\rho}$ and $N_w$, as shown in Fig. 8, show considerable similarities, suggesting a strong correlation.

The velocity-diameter fits shown represent a low  ensemble mean density ($\bar{\rho} = 0.05$ g cm$^{-3}$) time interval 01:03-01:16 (panel 8b) and two intervals 22:30-22:52 and 02:06-02:14 (8a, c) with higher  values of $\bar{\rho}$, 0.10 and 0.12 g cm$^{-3}$, respectively. Notable is the higher modal fall velocities and the absence of particles larger than 3 mm in the high density time intervals compared to the distribution in panel 8b.

**4.2 v-D and density**

In Fig. 9, particle fall velocity versus diameter data points combined from all the cases in Table 1 are divided into three categories according to the  snow ensemble mean density of the time interval during which particles were observed. A least squares fit is applied to observations in each  $\bar{\rho}$ range using the same procedure as for velocity dimensional fits for integration time intervals, as described in section 3.5. The total number of observed particles is roughly 4,440,000, and for each  density category numbers of particles included in the fitting process (within the red lines in Fig. 9) are approximately 1,140,000, 1,190,000 and 360,000, respectively. The fitted relations for  ensemble mean density ranges are

$$v(D) = 0.834D^{0.217}, \quad 0.0\,\text{g cm}^{-3} < \underline{\rho}\,\bar{\rho} \leq 0.1\,\text{g cm}^{-3}, \tag{11}$$

$$v(D) = 0.895D^{0.244}, \quad 0.1\,\text{g cm}^{-3} < \underline{\rho}\,\bar{\rho} \leq 0.2\,\text{g cm}^{-3} \text{ and} \tag{12}$$

$$v(D) = 0.906D^{0.256}, \quad \underline{\rho}\,\bar{\rho} \geq 0.2\,\text{g cm}^{-3}., \tag{13}$$

with RMSE values of 0.30 m s$^{-1}$, 0.30 m s$^{-1}$ and 0.35 m s$^{-1}$, respectively.

The coefficient is increased with density indicating higher fall velocities with more dense particles. There is also a clear increase in the slope of the fitted curve from the lowest  density range to the $0.1\ldots0.2$ g cm$^{-3}$ range indicated by the increase in the power term. With particles in the highest  density range the observed size distribution is narrow, hence the correlation between particle size and fall velocity is weak, and it is difficult to find an unambiguous relation between them. All things considered, the results are in line with the conclusion made by Barthazy and Schefold (2006), that the  prefactor and power terms increase with riming degree, which in turn is strongly connected with density (Power et al., 1964) .

Considering the definition of the volume equivalent diameter, relations in the form of (11)…(13) should be ideal for velocity-dimensional parametrization of radar observations as the average size of hydrometeors as observed by radar are largely defined by their volumes rather than their shapes.

**4.3 Connection between PSD parameters and density**

From the analysis of PSD parameters and their relations to  ensemble mean density we have excluded data points representing integration time intervals where $D_0 < 0.6$ mm, as lower values of median volume diameter would imply that a substantial fraction of particles are too small to be observed with PIP. Applying this restriction, along with minimum thresholds set for particle count and LWE precipitation rate in density retrievals, as described in section 3.2, all in all 101 time intervals were discarded from the total of 1141 intervals of observations, leaving 7173 minutes of snow observations for the analysis.

**4.3.1 Density and $D_0$**

In Fig. 10, observed distributions of $D_0$ for the three different density regimes are shown. For the low density particles, the maximum $D_0$ value  rarely exceeds 5 mm, which is in agreement with observations of snow aggregates presented by Lo and Passarelli (1982). It can also be seen that $D_0$ distribution depends on density. Low density particles are generally larger and vice versa. This dependence of $D_0$ on  ensemble mean density is not surprising, given that they are related as was previously shown by B07 and discussed in more detail below.

Relation between  $\bar{\rho}$ and size ($D_0$) is illustrated in Fig. 11. The areas of individual data points are proportional to the particle counts of the corresponding observation time intervals. The overlaid black solid curve, a least squares fit applied for all cases in Table 1 is given by

$$\rho\bar{\rho}(D_0) = 0.226 D_0^{-1.004}, \tag{14}$$

where $D_0$ is in millimeters and  $\bar{\rho}$ is in g cm$^{-3}$. As the two examined winters were seen to have notable differences between each other in the snowfall type and average  $\bar{\rho}$, corresponding relations were also calculated separately for the winters, and are given by

$$\rho\bar{\rho}(D_0) = 0.273 D_0^{-0.998} \quad \text{and} \tag{15}$$

$$\rho\bar{\rho}(D_0) = 0.209 D_0^{-0.969} \tag{16}$$

for BAECC events and for events of winter 2014/15, respectively. A relation by B07, given by  $\bar{\rho}(D) = 0.178 D_0^{-0.922}$, is plotted in Fig. 11 for comparison. As their definitions of particle diameter and  $\bar{\rho}$ are close to ours, the relations are easy to compare. Especially (16) is in good agreement with B07's results. The  ensemble mean density is on average higher for snow events recorded during BAECC, which suggests more riming occurred during those events. Indication to this is that the ARM AMF2 dual-channel microwave radiometer located on the same measurement field detected the presence of liquid water more than 80 % of the BAECC SNEX campaign time (Petäjä et al., 2016) and the presence of supercooled liquid layers could also be observed in the backscatter coefficient and circular depolarization ratio measurements of the co-located ARM HSRL (High Spectral Resolution Lidar) in the majority of the BAECC cases (Goldsmith et al., 2014). In general the BAECC winter was milder than the next winter 2014–2015, and the case duration weighted average of maximum recorded

temperatures was almost one degree higher for BAECC events compared to the value for winter 2014–2015 cases. The temperatures closer to 0°C could mean increased aggregation as stated in B07 and therefore decreased density values, but also different snow habits compared to  colder cases.

The mass-dimensional relation in power-law format $m = a_m D_m^b$  $m = a_m D^{b_m}$ can be derived from the retrieved  $\bar{\rho}$-$D_0$ relations (14) to (16) by assuming  gamma PSD and describing the  ensemble mean density approximately as

$$\rho\bar{\rho} = \frac{\int m(D)v(D)N(D)\mathrm{d}D}{\int V(D)v(D)N(D)\mathrm{d}D} \approx \frac{\int_0^\infty m(D)v(D)N(D)\mathrm{d}D}{\int_0^\infty V(D)v(D)N(D)\mathrm{d}D} \tag{17}$$

$$= \frac{\int a_m(D)^{b_m} a_v D^{b_v} N_0 \exp(-\Lambda D)\mathrm{d}D}{\int \frac{\pi}{6}(0.1D)^3 a_v D^{b_v} N_0 \exp(-\Lambda D)\mathrm{d}D} \frac{\int_0^\infty a_m(D)^{b_m} a_v D^{b_v} N_0 D^\mu \exp(-\Lambda D)\mathrm{d}D}{\int_0^\infty \frac{\pi}{6}(0.1D)^3 a_v D^{b_v} N_0 D^\mu \exp(-\Lambda D)\mathrm{d}D} \tag{18}$$

$$= \frac{6}{\pi}\frac{6}{\pi} 10^3 a_m \frac{\Gamma(b_m+b_v+1)}{\Gamma(4+b_v)} D_0^{b_m-3} \frac{\Gamma(b_m+b_v+\mu+1)}{\Gamma(b_v+\mu+4)} \left(\frac{0.1}{3.67}\frac{1}{3.67}\right)^{b_m-3} D_0^{b_m-3}. \tag{19}$$

The integration limits are defined from zero to infinity for deriving the analytic solution, though the true range is narrower because of left and right truncation of the observed size spectrum. As shown in Fig. 6, the ensemble mean density is overestimated because of the truncation. The estimation bias of density is ranging between 20 % for $D_0$ smaller than 0.75 mm and about 2 % for $D_0$ larger than 2 mm. Since for the estimation of the $m$-$D$ relation, most of the observed $D_0$ values are higher than approx. 1 mm as shown in Fig. 10, there is only minor contribution of the smaller $D_0$ values, and we are assuming our error in ensemble mean density because of truncation to be close to 2 %. This corresponds to an error of 2 % also in the prefactor $a_m$, if assumed that the truncation does not introduce significant changes in the exponents of the $\bar{\rho}$-$D_0$ and $m$-$D$ relations.

Taking the three velocity exponents from equations (11) to (13), and assuming exponential PSD, the derived prefactors and exponents of mass-relation  are shown in Table 2, having the volume-equivalent diameter proxy in mm and mass given in grams. The factor 0.1 in (18) derives from unit conversion, as  $\bar{\rho}$ is in g cm$^{-3}$. The values of prefactor $a_m$ are not sensitive to the changes in the velocity exponent $b_v$ (changes in $b_v$ are resulting less than 1 % devition $a_m$ values), though there is a small increase in $a_m$ with increasing $b_v$. The prefactor is more sensitive to shape parameter $\mu$ of the gamma PSD, the value of $a_m$ increases by 24 % as $\mu$ is increased from zero to 1. With value of $\mu = 3$ the increase in the prefactor $a_m$ value is 48 %. The shape factor of snow PSD is known to be noisy and thus often exponential distribution is assumed. With $b_v = 0.217$ the derived mass-dimensional relations for all cases and for both studied winters separately are plotted against literature values in Fig. 12. The derived exponent $b_m$ for the studied cases is in line with literature values, close to 2, but the prefactor $a_m$ values are higher than in the presented relations in Table 3. The highest value of $a_m$ is for the BAECC cases indicating conditions of riming. The high prefactor values might manifest the Finnish winter conditions, because of the vicinity of Baltic Sea, the air is more moist than e.g. in continental conditions.

**4.3.2 $N_w$ and density**

Distributions of observed $N_w$ values also exhibit dependence of $N_w$ on the ensemble mean density, as shown in Fig. 13, i.e. $N_w$ increases with density. The modal values of $N_w$ are approximately 5000, 40,000 and 80,000 $\mathrm{mm}^{-1}\mathrm{m}^{-3}$ for ensemble mean density ranges $0.0\ldots0.1$, $0.1\ldots0.2$ and $>0.2$ $\mathrm{g\,cm}^{-3}$, respectively, with vast majority of $N_w$ values spanning less than two orders of magnitude for a given $\bar{\rho}$ range. This dependence of $N_w$ on density is somewhat unexpected. There is no obvious reason to expect that $N_w$ would depend on density. However, because $D_0$ and density are related, dependence of $N_w$ on density potentially arises from the dependence of $N_w$ on $D_0$.

To verify this, the partial correlation analysis of the relation between log values of $N_w$ and density while controlling for log value of $D_0$ was carried out. It was found that there is a moderate negative partial correlation, -0.33, between $N_w$ and density while controlling for $D_0$. However, the zero-order correlation between $N_w$ and density is 0.52. The analysis confirms that the observed relation between $N_w$ and density, is due to their relation to $D_0$. It is not clear, however, what is the meaning of the found negative partial correlation between $N_w$ and density.

[revised manuscript text omitted]

*Acknowledgements.* We would like to acknowledge the Hyytiälä station and University of Helsinki personnel for the daily tasks with measurements, especially mentioning Matti Leskinen and Janne Levula. The research of JT and DM was supported by Academy of Finland (grant 263333) and the Academy of Finland Finnish Center of Excellence program (grant 272041). AvL was funded by grant of the Vilho, Yrjö 15 and Kalle Väisälä Foundation and by SESAR Joint Undertaking Horizon 2020 grant agreement No 699221 (PNOWWA). The instrumentation used in this study was supported by NASA Global Precipitation Measurement Mission ground validation program and by the Office of Science U.S. Department of Energy ARM program.

[Figure]

**Figure 1.** Snow precipitation instruments on the measurement field in Hyytiälä. The Pluvio$^2$ 200 is inside the wind protection on a platform and the PIP lamp can be seen at right on the ground. The view of the picture is to southwest and the distance from the platform to the treeline behind is approximately 20 m.

[Figure]

**Figure 2.**  The ratio of the diameter observed by PIP, $D_{PIP}$, to volume equivalent diameter $D$.

[Figure]

**Figure 3.** The distributions of snowflake a) aspect ratio and b) area ratio as observed using PIP with interquartile ranges visualized and median values shown.

[Figure]

**Figure 4.** Distribution of snow ratios, ratio of snow depth change to LWE, calculated from retrieved ensemble mean densities with interquartile range, and median and mean values.

[Figure]

**Figure 5.** Scatterplot of the hourly change of snow depth measured with Jenoptik SMH30 and estimated from volume flux using PSD and fall velocities as measured by PIP. The data includes all the studied cases except Jan 10-11 2015.

[Figure]

**Figure 6.** Computed normalized bias and standard deviation of estimated mean snow density as a function of $\mu$ and $D_0$. The shaded area indicates data that is not included in the analysis, because derived $D_0$ is smaller than 0.6 mm. The increased values of bias at low $D_0$ values is due to left side truncation of the observed PSD, which is caused by the instrument sensitivity. At larger $D_0$ values the bias approaches value of 0.02.

[Figure]

**Figure 7.** Evolution of snowfall intensity,  ensemble mean density and particle size distribution parameters during March 18th 2015 with associated ($v$, $D$) from three selected time intervals (highlighted in gray). The red dashed lines mark the upper and lower velocity limits where for a given $D$, the KDE value is higher than half maximum.

[Figure]

**Figure 8.** Evolution of snowfall intensity,  ensemble mean density and particle size distribution parameters during the night between the 22nd and 23rd of January 2015 with associated $(v, D)$ from three selected time intervals (highlighted in grey). The red dashed lines mark the upper and lower velocity limits where for a given $D$, the KDE value is higher than half maximum.

[Figure]

**Figure 9.**  Probability densities of $(D, v)$ in three  ensemble mean density ranges ($[\bar\rho] = \mathrm{g\,cm}^{-3}$). Dashed lines mark the full width at half maximum KDE in each diameter bin. Power law functions are fitted for data between those lines.

[Figure]

**Figure 10.** Normalized frequency (bars) and kernel density (line) of median volume diameter $D_0$ in three  ensemble mean density ranges, $[\bar{\rho}] = \mathrm{g\,cm}^{-3}$.

[Figure]

**Figure 11.** ($D_0$,  $\bar{\rho}$) for all cases listed in  Table 1. Area of each dot is proportional to the number of particles in corresponding integration time interval. Power law fits are shown separately for BAECC winter cases (blue) and cases from the following winter (green).

[Figure]

**Figure 12.** Derived $m$-$D$ relations assuming exponential PSD in comparison relations presented in literature are shown in Table 3.  A conversion of maximum dimension to volume equivalent diameter is done by assuming axis ratio of 0.6.

[Figure]

**Figure 13.** Frequency of $N_w$ in three  ensemble mean density ranges,  $[\bar{\rho}]$ = g cm  $^{-3}$.

[Figure]

**Figure 14.** $(D_0, N_w)$ and ($\bcancel{D_0\rho^{1/3}}\underbrace{D_0\bar{\rho}^{1/3}}$, $N_w$) with fitted relations

[Figure]

**Figure 15.** Normalized frequency (bars) and kernel density (line) of the gamma PSD shape factor $\mu$ in three  ensemble mean density ranges,  $[\bar{\rho}]$ = g cm$^{-3}$.

**Table 1.** Liquid water equivalent precipitation accumulation measured with Pluvio[2] 200 and 400, change in snow depth and maximum and minimum temperature,maximum and minimum relative humidity, mean and maximum wind speed and mean wind direction of the studied snow events. Events before the horizontal line are recorded during the BAECC campaign.

| Event | LWE (mm) 200 | LWE (mm) 400 | $\Delta$SD (cm) | Temp (°C) min | Temp (°C) max | RH (%) min | RH (%) max | Wind (m s$^{-1}$, °) mean | Wind (m s$^{-1}$, °) max | Wind (m s$^{-1}$, °) mean dir. |
|---|---|---|---|---|---|---|---|---|---|---|
| 2014 Jan 31 21:00 - Feb 01 06:00 | 7.4 | 7.3 | 5.1 | -9.8 | -8.9 | 84 | 91 | 1.6 | 2.9 | 138 |
| 2014 Feb 12 04:00 - 11:00 | 1.0 | 0.9 | 1.8 | - 1 | 0 | 96 | 98 | 0.6 | 2.0 | 170 |
| 2014 Feb 15 21:00 - Feb 16 03:00 | 2.6 | 2.6 | 2.5 | -2.1 | -1 | 86 | 97 | 1.9 | 2.7 | 140 |
| 2014 Feb 21 16:00 - Feb 22 05:00 | 5.5 | 5.2 | 3.6 | -2.7 | 0 | 88 | 98 | 2.1 | 3.4 | 138 |
| 2014 Mar 18 08:00 - 19:00 | 4.4 | 4.0 | 7.3 | - 3.8 | -1.8 | 76 | 96 | 1.2 | 2.7 | 155 |
| 2014 Mar 20 16:00 - 23:00 | 6.1 | 5.9 | 4.8 | - 4.3 | -1.3 | 89 | 97 | 2.0 | 3.4 | 146 |
| 2014 Nov 06 19:00 - Nov 07 14:30 | 10.5 | – | 10.3 | -2.4 | -1.6 | 95 | 97 | 0.8 | 1.9 | 238 |
| 2014 Dec 18 14:00 - 19:00 | 2.6 | 2.2 | 3.9 | -2.3 | -0.8 | 97 | 98 | 1.0 | 1.8 | 134 |
| 2014 Dec 24 08:30 - 13:00 | 1.3 | 1.2 | 1.2 | -9.2 | -8.9 | 90 | 91 | 0.7 | 1.5 | 204 |
| 2014 Dec 30 00:30 - 14:00 | 6.3 | 5.3 | 4.9 | -10.4 | -0.6 | 91 | 98 | – | – | – |
| 2015 Jan 3 09:00 - 23:50 | 7.3 | 7.3 | 11.9 | - 3.9 | 0 | 96 | 98 | 2.6 | 5.2 | 318 |
| 2015 Jan 7 01:00 - 20:10 | 5.4 | 4.8 | 2.2 | -6.5 | -0.8 | 92 | 97 | 1.3 | 2.8 | 181 |
| 2015 Jan 8 06:00 - 13:30 | 2.6 | 2.7 | 1.6 | - 1.9 | 0 | 97 | 99 | 1.0 | 2.2 | 155 |
| 2015 Jan 9 18:00 - Jan 10 06:00 | 3.1 | 3.1 | 4.6 | -3.7 | -0.2 | 95 | 98 | 1.0 | 3.0 | 286 |
| 2015 Jan 10 22:00 - Jan 11 09:00 | 0.7 | 0.6 | 0.7 | -12.6 | -4.4 | 88 | 95 | 1.6 | 3.4 | 207 |
| 2015 Jan 12 21:00 - Jan 13 08:30 | 12.8 | 10.9 | 9.6 | -15.7 | -9.0 | 88 | 94 | 1.3 | 3.1 | 181 |
| 2015 Jan 13 22:00 - Jan 14 07:00 | –* | 2.2 | 1.9 | -8.0 | -0.3 | 94 | 98 | 0.5 | 1.9 | 134 |
| 2015 Jan 16 01:30 - 07:30 | –* | 5.8 | 5.2 | -1.3 | -0.6 | 92 | 98 | 1.9 | 3.4 | 154 |
| 2015 Jan 18 16:00 - 21:00 | 1.9 | 1.9 | 2.7 | -2.4 | -0.3 | 95 | 97 | 1.2 | 2.6 | 300 |
| 2015 Jan 22 21:00 - Jan 23 04:30 | 2.1 | 2.0 | 2.3 | -13.3 | -12.5 | 87 | 90 | – | – | – |
| 2015 Jan 23 15:00 - 23:00 | 1.4 | 1.2 | 1.4 | -10.1 | -8.8 | 91 | 93 | 0.3 | 1.0 | 205 |
| 2015 Jan 25 09:00 - 16:00 | 2.8 | 2.5 | 1.9 | -2.4 | -1.7 | 96 | 97 | 0.7 | 1.7 | 170 |
| 2015 Jan 31 12:00 - Jan 31 23:15 | 7.0 | 6.6 | 5.7 | -1.9 | -0.4 | 92 | 97 | 1.2 | 2.6 | 175 |

*Pluvio[2] 400 was used as data from Pluvio[2] 200 was unavailable

**Table 2.** The prefactors and exponents of $m = a_m D^{b_m}$ derived for exponential PSD with different values of exponent $b_v$ of velocity relation. The mass given in grams and the volume-equivalent diameter proxy in mm.

| Dataset | $b_m$ | $a_m(b_v = 0.217)$ | $a_m(b_v = 0.244)$ | $a_m(b_v = 0.256)$ |
|---|---|---|---|---|
| All cases | 1.996 | 1.036e-4 | 1.045e-4 | 1.049e-4 |
| BAECC cases | 2.002 | 1.254e-4 | 1.264e-4 | 1.269e-4 |
| Winter 2014-2015 cases | 2.031 | 9.679e-5 | 9.757e-5 | 9.792e-5 |

**Table 3.** The prefactors and exponents of $m = a_m D^{b_m}$ of literature values for comparison plotted in Fig. 12. A conversion from maximum dimension to volume equivalent diameter is done by assuming axis ratio of 0.6.

| Study | $b_m$ | $a_m$ |
|---|---|---|
| Matrosov 2007Matrosov 2007 , $0.12\,\mathrm{mm} < D \leq 2.4\,\mathrm{mm}$ | 2.0 | $4.2172 \times 10^{-5}$ |
| Matrosov 2007Matrosov 2007 , $2.4\,\mathrm{mm} < D \leq 24\,\mathrm{mm}$ | 2.5 | $3.2430 \times 10^{-5}$ |
| Heymsfield et al. 2004 Heymsfield et al. 2004 | 2.04 | $7.5814 \times 10^{-5}$ |
| Mitchell et al. 1990 Mitchell et al. 1990 | 2.0 | $3.0926 \times 10^{-5}$ |
| Locatelli and Hobbs 1974 Locatelli and Hobbs 1974 | 1.9 | $5.1134 \times 10^{-5}$ |

---

## Author Comment (AC2) · 26 Aug 2016

The authors deeply appreciate the time and efforts of the Anonymous referee #3 in reviewing the manuscript. We have addressed each of your concerns in our response below.

A marked-up version of the manuscript indicating the changes we have made is attached as a supplement to this response. All page and line numbers in our comments are in reference to that document.

[Figure]

**1   General comments**

A small correction to your summary of the manuscript: Instead of using a "5-min (or more) temporal resolution", the variable temporal resolution for derivation of mean snow bulk density and PSD parameters could go down to a minimum of 1 minute (in the case that LWE precipitation intensity was higher than 0.1 mm min$^{-1}$). The shortest integration time, $\tau(t)$, in the analysis of cases in Table 1 was two minutes. The following sentence was added to section 3.2 to clarify this:

*Effectively, the temporal resolution of the mean bulk density retrieval is increased with increasing precipitation intensity, and in the analysis of the snowfall events in Table 1, the median $\tau(t)$ was 5 minutes.*

***Q1**.   The equi-volume diameter of each individual snowflake/particle is estimated by multiplying the PIP equivalent-area diameter by 0.92. This number has been found by simulating the relationship between the PIP diameter of the 2D projection of a rotated spheroid and its equi-volume diameter (see p.5, l.16-21). This equi-volume diameter is central to the study as the PSD is derived from it, but there is no discussion about the uncertainty of this estimation. Real snowflakes are not rotated spheroids, and I am hence wondering what is the spread of the real equi-volume diameter around the 0.92 estimate, and subsequently the uncertainty in the fitted PSD parameters.*

**Response**. We have added a figure that shows how this correction factor depends on ellipsoid dimensions, assuming that snowflake shape can be approximated as an ellipsoid. We should note that this approximation is only used to capture main effects of viewing geometry on estimated particle size. Typically, the multiplication factor varies between 0.8 and 1. For ice particles with axis ratios smaller than 0.4, i.e. pristine ice crystals, this factor could approach 1.4. From this analysis we can see that the largest expected error is associated with observations of ice crystals. Dimensions of snowflake aggregates and graupel like particles are expected to be captured with a smaller error.

[Figure]

**Changes**. Added Fig. 2, discussion starting p.6 l.15.

*Q2. The relationship between $D_0$, $N_w$ and the density is quantified using linear correlation. This correlation quantifies the co-fluctuation of two random variables, but does not tell anything about causality. I would suggest to investigate the possible link between $N_w$ and density by using multiple or partial correlation, in order to remove the influence of $D_0$ in the co-fluctuation of $N_w$ and density.*

**Response**. Thank you for the suggestion. We have performed the partial correlation analysis of the relation between $N_w$ and density while controlling for $D_0$. It was found that there is a moderate negative partial correlation, -0.33, between $N_w$ and density while controlling for $D_0$. However, the zero-order correlation between $N_w$ and density is 0.52. The analysis confirms that the observed relation between $N_w$ and density, is due to their relation to $D_0$. It is not clear, however, what is the meaning of the found negative partial correlation between $N_w$ and density. We have added the corresponding text to the manuscript.

**Changes**. Added a paragraph starting p.15 l.12.

*Q3. It is more a wish than a request: the PIP provides information about the particle type, and it would be very interesting to conditioned the analysis on particle type as well...*

**Response**. Indeed, it would be a very interesting study. As there currently exists no reliable automatic classification method for particle type for the PIP data, such analysis is out of the scope of this study. The PIP provides information on particle shape, which is currently not used for automatic classification.

[Figure]

**2 Specific and technical comments**

*Q1. Title: I would change the word "snow" into "snowfall", to clearly indicate the difference with studies of snow density on the ground.*

**Response**. Changed to "falling snow".

*Q2. P.2, l.8: Garrett et al. (2012) would be a better reference here I think.*

**Response**. Corrected.

*Q3. P.6, l.9: a reference about the employed method of moments?*

**Response**. Added.

*Q4. P.6, l.26: correspondingly.*

**Response**. Corrected.

*Q5. P.6, l.27: How many time steps are filtered here, and how representative is the remaining set?*

**Response**. The filter conditions are summarized, and the total numbers of time steps included and excluded in the analysis given in the beginning of section 4.3 (p.13 l.9-11).

*Q6. P.7, Eq.8: a $D_{max}$ is provided as upper integration limit, while it is implicitly assumed to be $+\infty$ in Eq. 2-3. Please clarify.*

**Response**. We have added an additional error analysis to see how this affects our retrieval. The results of this analysis are summarized in the new Fig. 6.

**Changes**. Added a new Chapter 3.4 (starting p.9 l.35).

*Q7*. *P.7, l.27: the mean/median values are close to the commonly assumed values, but Fig. 2 shows a potentially significant spread around these values. For instance, the mode (most likely value) is around 6-7, and it would be instructive to provide an interquantile range (asymmetric distribution) to quantify this spread.*

**Response**. We have added interquantile ranges to the Figs. 3 - 5.

*Q8*. *P.8, l.3: "the agreement is rather good": please provide quantitative descriptors (correlation, RMSE, bias, etc.) of this agreement.*

**Response**. We have given quantified measures for the comparison and the text has been altered:

**Changes**. "It can be seen that the agreement is good, with RMSE of 0.30 cm, linear correlation coefficient of 0.88 and normalized bias as low as $-0.06$." (p.9 l.32)

*Q9*. *P.8, l.10-11: fitting a power law in the log-log space using a linear regression does not provide optimum parameter values in the linear space... It should be mentioned.*

**Response**. This point is now stated and justified by the fact that we don't want the fits to overly emphasize the large end of PSD.

**Changes**. At the end of p.10: "It should be noted, that using linear regression in log-log space does not optimally minimize residuals in linear space, but the method is used here as it does not overly emphasize the large end of the size spectrum."

*Q10*. *P.8, l. 29: I would move "were recorded" in between "rates" and "on average".*

**Response**. Corrected.

***Q11****. P.9, l.5: any comment on the possible explanations of this variability?*

**Response**. It is probably associated with riming. higher $a_v$ and $b_v$ values correspond to rimed particles.

**Changes**. Addition to p.11 l.32: "–, which possibly indicates the onset of riming."

***Q12****. P.9, Eq. 11-13: please provide a quantitative descriptor of the goodness-of-fit of these power laws!*

**Response**. Added values of RMSE.

**Changes**. Addition to p.12 l.26: "with RMSE values of 0.30 m s$^{-1}$, 0.30 m s$^{-1}$ and 0.35 m s$^{-1}$, respectively."

***Q13****. P.10, l.3: the term "riming degree" is coming out of the blue here...*

**Response**. The link between riming and density is now stated in this context.

**Changes**. Addition to p.12 l.32: "–, which in turn are strongly connected with density (Power et. al. 1964)."

***Q14****. P.11, l.5: remove "more" before "colder".*

**Response**. Corrected.

***Q15****. P.11, Eq.17-18: please specify what are the integration limits! In Eq.18, shouldn't it be $0.1^{-3}$ rather than $0.1^{b_m-3}$?*

**Response**. Integration limits have been added with discussion after the equations. Additionally, there was an error in (19), it should have been $0.1^{-3}$, which is now also

corrected. While errors were present in the equations in the manuscript, the values have been calculated correctly.

**Changes**. Corrected equations starting on p.14 l.17 and added supplementary discussion.

*Q16. Fig.4 and 5: it is not easy to spot the a, b, c markers. They should be made more visible (upper part?).*

**Response**. Thanks for the note and suggestion. Markers have been moved up and the corresponding time intervals have been highlighted with light grey background colour.

*Q17. Fig.12: if the minimum integration limit is $D = 0$, then $\mu$ values should be strictly positive (otherwise N(D) is not defined). But there are values down to -2 in Fig. 12. . .*

**Response**. This is a known problem in the PSD parameter estimation. $\mu$ values should be larger than -1. Values below or equal to -1 would result in ill defined total number concentrations, for example, if calculated from such a distribution. However, because of estimation errors and because actual PSD are not necessary following the Gamma functional form, sometimes $\mu$ values found to be smaller or equal to -1. This is happening because when we estimate $N_w$, $D_0$ and $\mu$, we are looking for a Gamma function that fits the best to an observed PSD. This Gamma function may not be in a strict sense a PSD, because we cannot calculate an $N_t$ from the fitted parameters, for example. As you probably noticed, we are not using derived $\mu$ values quantitatively. We just concluded that the observed values are close to 0.

**Supplement:**

**Ensemble mean density and its connection to other microphysical properties of falling snow as observed in Southern Finland**

[revised manuscript text omitted]

15  By combining optical disdrometer observations with other measurements, e.g. by radar or precipitation gauge, physical properties such as mean snow density can be derived. Huang et al. (2010) have used a C-band weather radar observations of equivalent reflectivity factor, $Z_e$, in combination with a 2DVD to derive snow density-dimensional relation and to infer more consistent $Z_e$–snowfall rate, SR, relations. Another method for snow density retrieval is based on solving aerodynamic equations to derive particle mass from observed fall velocity and particle effective projected area as proposed
20  by Böhm (1989) and applied by Hanesch (1999) and more recently by Szyrmer and Zawadzki (2010) and Huang et al. (2015) . Brandes et al. (2007), hereafter referred to as B07, used a combination of a weighing gauge and a 2DVD to derive mean bulk density-median volume diameter relations and to document relations between PSD parameters for Colorado winter storms. Their approach is similar to the one used by Heymsfield et al. (2004) who have combined aircraft PSD and ice water content observations to derive  mean snow density and average mass-dimensional relations for ice particles. Albeit using slightly
25  different definitions, both B07 and Heymsfield et al. (2004) derive effective ice densities for ensembles of ice particles, but there is a difference in terminology.  Heymsfield et al. (2004) and many others have used the term (particle) bulk density to refer to the
30   density of individual ice or snow particles defined as the ratio of mass of a particle having a size $D$ to its assumed volume: $\rho = \rho(D)$. In most of such cases, the word "bulk" is used to emphasize the inclusion of hollows within particles. The term "(mean) bulk density" is sometimes used also when referring to the mean density of an ensemble of particles representing the whole PSD, i.e. $\bar{\rho} = \bar{\rho}(D_0)$ (e.g., B07) , whereas Heymsfield et al. (2004) used the term population-mean effective density. In this study we

derive the volume flux weighted snow density, similar to e.g. B07 , and refer to it as ensemble mean density, $\bar{\rho}$, to avoid possible confusion.

This paper documents connection between  ensemble mean density and other microphysical properties of snow as observed in Southern Finland. Using the estimated $\bar{\rho}$, average mass-dimensional relations characteristic to studied snowfall events are defined. In order to derive  ensemble mean density, a method proposed by B07 was used. However, instead of a 2DVD, a new generation of the SVI is employed. It is shown that, despite simpler construction compared to the 2DVD, this instrument's data is suitable for such studies.

Even though this study is based on retrieval of ensemble mean snow density and not mass-dimensional relations directly, which could be more easily applied to radar retrievals and numerical weather prediction (NWP). There are a number of applications of such relations. Aikins et al. (2016) used $\bar{\rho}(D_0)$ to convert particle size distribution observations to precipitation rate. Tong and Xue (2008); Dolan and Rutledge (2009); Matrosov et al. (2009); Huang et al. (2010); Zhang et al. (2011) used mean snow density-median volume diameter relations for characterizing winter precipitation microphysics by radar. Kneifel et al. (2015) showed a connection between mean snow density and multi-frequency radar observations. Thompson et al. (2008) used the density relation by B07 , and Iguchi et al. (2012) applied a similar density retrieval method to improve parametrization of snow microphyics in NWP models, for example.

**2   Measurements**

**2.1   Measurement setup**

[revised manuscript text omitted]

**2.4 Snow depth sensor**

The laser snow depth sensor, Jenoptik SHM30, is located on the measurement field, next to Pluvio$^2$ 400. It is an  optical sensor, which measures the snow depth by comparing signal phase information of the modulated visible laser light. It is a point measurement, hence the piling of wind driven snow or random branches and leaves drifting on the snow pack can cause misreadings. To reduce this we have sheltered the measurement spot with a small wind fence and the instrument structure excluding the measurement pole is buried under the ground to prevent the piling of snow. The data is recorded every minute.

**3 Retrievals of  ensemble mean density, velocity-dimensional relations and PSD**

Observations from the PIP and one of the weighing gauges are combined to retrieve  snow ensemble mean density. Typically the gauge located inside of the DFIR, the Pluvio$^2$ 200, is used for this retrieval. On a couple of days this gauge was not operational and data from the Pluvio$^2$ 400  located outside of the DFIR was used instead. These dates are marked in Table 1 with asterisks in the LWE precipitation rate column. As seen in the Table 1 the differences in accumulated LWE recorded by the two Pluvio$^2$s are small, largest being 15 %. Pluvio$^2$ 200 inside the DFIR is typically measuring higher accumulations, which is expected because of the  better wind protection. However, the observations do not show a clear indication that  the observed precipitaiton accumulation difference depends on the wind speed. However, the difference seems to increase in respect to certain wind directions. There are two openings from the measurement field, one to a road crossing (approx. 130°) and the other to small field (approx. 180°). If the wind is blowing from these directions the difference between the two gauges seem to increase.

The retrieval procedure is described below and is similar to the one presented by B07, but with notable modifications. Prior to retrieval of $\bar{\rho}$, PSD and velocity-dimensional relations are estimated. It was found, however, that the density retrieval is highly sensitive to the integration time. To minimize this, a variable integraiton time determined by the precipitation accumulation is used. The same integration time was applied to compute PSD parameters and $v$-$D$ relations.

**3.1 Particle size distribution**

The PSDs are calculated from the PIP records of particles that fell through the observation volume. The observed distributions are defined with respect to equivalent area diameter $D_{PIP}$, which is different from the apparent diameter of the 2DVD

5  and maximum particle dimensions used in other studies (e.g., Heymsfield et al., 2004). Wood et al. (2013) studied differences between diameter definitions and found that the diameter recorded by SVI is approximately 0.82 of maximum particle dimension. We performed a similar study by examining mean dimensions of  rotated ellipsoids on a single projection, as shown in Fig. 2. The ellipsoids were defined by a long dimension $a$ and a short dimension $b$ lying nominally in the horizontal plane along the $x$ and  $y$-axes, respectively, and

10  a short vertical dimension $c$ lying nominally along the $z$-axis. The particle orientation was defined by Gaussian distribution of canting angles with a standard deviation of $9°$ (Matrosov et al., 2005a) and a uniform distribution of azimuth angles. The equivalent area diameters $D_{\mathrm{PIP}}$ of simulated particles were estimated as their projected areas onto the $x$-$z$ plane and the resulting values were averaged over all orientations. The ratios of mean $D_{\mathrm{PIP}}$ to the particle volume equivalent diameter, i.e. the diameter for which the particle volume $V(D) = \frac{\pi}{6}D^3$

15  , for a number of combinations of vertical and horizontal aspect ratios are shown in Fig. 2. Assuming spheroids (Matrosov, 2007) and taking the typical vertical aspect ratio $c/a = 0.6$ (Korolev and Isaac, 2003; Matrosov et al., 2005b) we found that $D_{\mathrm{PIP}}$ is roughly equal to 0.92 of a volume equivalent diameter. As can be seen, the conversion factor varies between 0.8 and 1. For ice particles with axis ratios smaller than 0.4, i.e. pristine ice crystals, this factor could approach 1.4. From this analysis we can conclude that the largest expected error is associated with observations of ice crystals. Dimensions

20  of snowflake aggregates and graupel like particles are expected to be captured with a smaller error. In this study the same conversion factor of  0.92 is used for all the cases. As can be seen in Fig. 3 the median area and aspect ratios of the particles are 0.65 and  0.72, respectively. These observations also support our choice of a mean particle shape and the corresponding

25  diameter transformation. Therefore, the results presented in the rest of the manuscript are using this volume equivalent diameter proxy.

Prior to calculations of PSD parameters, recorded PSD data is filtered to remove spurious observations of large particles. Following the procedure described in  ?, records of large particles were ignored if there was a gap of more than three consecutive PSD diameter bins. The bin size was set to 0.25 mm during the BAECC experiment and it was reduced to 0.2 mm for the winter 2014/2015. The PIP resolution is 0.1 mm and the minimum detectable particle diameter is approximately 0.3 mm (Newman et al., 2009). The smallest diameter bin used in calculations is 0.25 mm to 0.5 mm during BAECC and 0.2 mm to 0.4 mm in the following winter.

5  The PSD parameters were calculated using method of moments and assuming that PSD follows gamma functional form, see for example Ulbrich and Atlas (1998) and citations therein. The normalized gamma distribution $N(D)$ in $\mathrm{mm^{-1}m^{-3}}$ was adopted following Testud et al. (2001); Bringi and Chandrasekar (2001); Illingworth and Blackman (2002):

$$N(D) = N_w f(\mu) \left(\frac{D}{D_0}\right)^\mu \exp(-\Lambda D) \tag{1}$$

$$f(\mu) = \frac{6}{3.67^4} \frac{(3.67+\mu)^{\mu+4}}{\Gamma(\mu+4)} \tag{2}$$

$$\Lambda = \frac{3.67+\mu}{D_0}, \tag{3}$$

with $N_w$ in $\mathrm{mm}^{-1}\mathrm{m}^{-3}$ being the intercept parameter, $D_0$ the median volume diameter in mm, $\Lambda$ the slope parameter in $\mathrm{mm}^{-1}$ and $\mu$ the shape parameter. Using the second, fourth and sixth moments for the non-truncated gamma PSD, $M_2$, $M_4$, and $M_6$, the PSD parameters were estimated as follows:

$$\eta = \frac{M_4^2}{M_6 M_2} \tag{4}$$

$$\mu = \frac{7 - 11\eta - \sqrt{\eta^2 + 14\eta + 1}}{2(\eta - 1)} \tag{5}$$

$$\Lambda = \sqrt{\frac{M_2 \Gamma(\mu+5)}{M_4 \Gamma(\mu+3)}} \tag{6}$$

$$D_0 = \frac{3.67 + \mu}{\Lambda} \tag{7}$$

**3.2  Ensemble mean density retrieval**

The integration time, $\tau(t)$, of the  ensemble mean density retrieval is driven by precipitation measurements of the Pluvio[2]. The step of the non-real-time accumulation output is 0.05 mm, causing the output interval to be in the order of several minutes even at moderate snow rates. With a short fixed integration time in time scales of minutes or tens of minutes, the produced  ensemble mean density estimation would hence be the more unstable, the lower the precipitation rate. Therefore, variable length time intervals driven by the gauge output are used with a selected threshold value of 0.1 mm. This corresponds to a $\tau(t)$ of 6 minutes for a LWE precipitation intensity of $1\,\mathrm{mm\,h}^{-1}$. Effectively, the temporal resolution of the ensemble mean density retrieval is increased with increasing precipitation intensity, and in the analysis of the snowfall events in Table 1, the median $\tau(t)$ was 5 minutes.

As the integration time  $\tau(t)$ is effectively driven by precipitation intensity, there is less variation in number of particles between intervals, compared to a fixed time interval approach. With the selected accumulation threshold there are typically between $10^3$ and $10^4$ particles within a given integration time interval. On the other hand, with low precipitation intensities, $\tau(t)$ increases up to one hour and retrieved  $\bar{\rho}$ becomes less representative for the time interval in question. With LWE precipitation rates lower than $0.2\,\mathrm{mm\,h}^{-1}$, the resolution of Pluvio[2] LWE measurements is insufficient and calculations of  $\bar{\rho}$ become overly sensitive to recorded number concentrations. Correspondingly, similar unwanted sensitivity to LWE precipitation accumulation occurs when the number of particles observed by PIP within $\tau(t)$ is less than 800. Therefore, time intervals with precipitation rates or particle counts lower than these thresholds are excluded from our analysis.

Given a population of solid precipitation particles with volume equivalent diameters $D$ over the integration time $\tau(t)$, the liquid equivalent precipitation accumulation in mm is approximately

$$G(t) \approx \frac{\pi}{6} \times 10^{-6} \frac{\bar{\rho}}{\rho_w} \int_t^{t+\tau(t)} \int_{D_{min}}^{D_{max}} D^3 v(D,t) N(D,t) \, \mathrm{d}D \, \mathrm{d}t, \tag{8}$$

10  where $\bar{\rho}$ is the volume flux weighted population mean snow density in $\mathrm{g\,cm^{-3}}$, $\rho_w = 1\ \mathrm{g\,cm^{-3}}$ is the density of liquid water, $N(D,t)$ is mean particle number concentration over the integration time in $\mathrm{mm^{-1}m^{-3}}$, $v(D,t)$ is particle velocity relation in $\mathrm{m\,s^{-1}}$ and $[D_{min}, D_{max}]$ is the size range of snowflake observations from a disdrometer. From (8) we can estimate volume flux weighted snow density for each observation time interval as

$$\bar{\rho}(t) \approx \frac{6}{\pi} \times 10^6 \rho_w \frac{G(t)}{\int_t^{t+\tau(t)} \int_{D_{min}}^{D_{max}} D^3 v(D,t) N(D,t) \, \mathrm{d}D \, \mathrm{d}t}, \tag{9}$$

15  using liquid equivalent precipitation accumulation $G(t)$ as measured by the Pluvio² gauge, and retrieving volume flux with fitted $v(D,t)$, and averaged $N(D,t)$ as measured by the PIP. It should be noted, that unlike in the retrieval of PSD parameters, where gamma PSD was assumed, $\bar{\rho}$ was retrieved without making any assumptions on the shape of the PSD distribution, and instead, measured PSD are used in the calculations.

**3.3  Comparison of derived mean density to snow depth observations**

20  The definition of ensemble mean density here is the same as for mean bulk density in B07. They determine the densities for 5-minute precipitation volumes derived with a 2DVD disdrometer observations together with precipitation mass measured by a weighing gauge. B07 defined the volume of a single particle by summing coin-shaped sub-volumes together estimated separately for both orthogonal projections and taking geometrical mean. As the diameter used in our study is the estimated volume-equivalent diameter, our results are comparable to B07. In Heymsfield et al. (2004), the volume of a

25  single particle is defined as a function of circumscribing maximum diameter, and the population mean effective density is determined from ice water content (IWC). The estimated ensemble mean snow density is volume-weighted and expected to have lower values than the velocity-weighted snow density. The difference is not generally prominent especially with low-density aggregates, whose velocity-dimensional dependence is weak.

It should be noted that the derived density is inversely proportional to the snow ratio, $R_s$, assuming that issues related

30  to packing of snowflakes on the ground can be ignored. The snow ratio (Power et al., 1964; Ware et al., 2006) is used by operational weather services to estimate change in snow depth from LWE observations and can be defined as follows:

$$R_s(t) = \frac{1}{P \cdot C} \frac{\rho_w}{\bar{\rho}(t)} \tag{10}$$

where $\bar{\rho}(t)$ is the volume flux weighted snow density derived as shown in (9), $P$ is the packing efficiency of snowflakes and $C$ is the snow compression. Assuming that the packing and compression terms, or their product

are close to unity, the derived density can be  tested against the commonly used assumption that 1 mm of LWE accumulation corresponds to 1 cm change in snow depth. In Fig. 4 the combined distribution of estimated snow ratios on temporal scales defined by the gauge accumulation for all the 23 events analyzed in this study is shown. It can be seen that the mean and median values, equal to 10 and 9 respectively, are very close to the commonly assumed value.

This analysis assumes that packing efficiency of snowflakes is 100 % and compression of snow on the ground can be ignored, or snow compression counteracts reduction in snow density due to packing. The packing efficiency of snowflakes on the ground is not known. Random packing of the same size spheres has density of 64 % and dense packing of such spheres uses 74 % of the volume, corresponding to $P = 0.64$ and 0.74 respectively. Packing efficiency of equal spheroids depends on axis ratios and exceeds this of spheres and could exceed 77 % (Donev et al., 2004) . It is not unreasonable to expect that irregular shaped particles of variable sizes, such as snowflakes, would pack more efficiently than equal spheroids. At least, packing efficiency in excess of 90 % can be expected for spheres of several radii (de Laat et al., 2014) . The packing efficiency of 70 % would mean that density of freshly fallen snow would be 30 % lower than this of falling snowflakes. The packing efficiency of 80 % would correspond to 20 % bias in estimated snowflake density from snow depth measurements, or in 25 % underestimation of the snow depth change by using $\bar{\rho}(t)$. We don't know the exact value of the snow packing, but could expect that in the worst case scenario it is about 70 % and probably closer to 80 % or even higher. It should also be noted that the snow compression would counteract this, but we are considering only freshly fallen snow and expect that the compression factor $C$ is very close to unity.

One of the major uncertainties in the density retrieval is the assumption about particle volume. In this study we have assumed that snowflakes are spheroids with axis ratios of 0.6. Given this assumption, a conversion factor relating volume equivalent and observed disc equivalent diameters was defined.  Fig. 2 shows that for a reasonable range of ellipsoid axis ratios this conversion factor can range between 0.8 and 1. This range of values implies that the uncertainty in the density estimation can range from an overestimation by as much as 50 % to an underestimation by about 20 %. This range of uncertainty is much larger than what is expected from a comparison of the retrieved volume-flux weighted density and snow depth measurements, as was discussed above. Therefore, by comparing the PIP derived and the directly measured snow depths, the validity of the derived  values of $\bar{\rho}$, and assumption of particle shape, can be checked. In Fig. 5 hourly change in the snow depth measured by the Jenoptik SHD30 is compared to the PIP derived snow depth. It can be seen that the agreement is   good, with RMSE of 0.30 cm, linear correlation coefficient of 0.88 and normalized bias as low as $-0.06$. This comparison also gives confidence about the validity of the derived  ensemble mean densities.

**3.4    Effect of PSD truncation on derived ensemble mean snow density**

The observed PSD are truncated on left and right sides (Ulbrich and Atlas, 1998) . They are truncated on the right side because of the instrument finite sampling volume and because natural sizes of hydrometeors do not extend to infinity. The truncation on the left, on the small-diameter side, is due to instrumental limitations and possible wind effects (Moisseev and Chandrasekar, 2007) .

Ulbrich and Atlas (1998) have presented a comprehensive analysis on how the right-side truncation affects the derived Gamma PSD parameters. A similar study on the effects of the left-side truncation and other instrumental effects was presented by Moisseev and Chandrasekar (2007) . Here we apply the method presented by Moisseev and Chandrasekar (2007) to estimate impact of PSD truncation on the derived mean snow density.

To investigate the impact of the PSD truncation on retrieval of mean snow density a simulation study was performed. To initiate the simulation the PSD parameters $N_w$, $D_0$ and $\mu$, together with parameters of $m$-$D$ and $v$-$D$ are used. During the study it was found that the density estimation error is most sensitive to $D_0$ and $\mu$ and virtually independent of the other input parameters. Therefore, results presented here assume that $N_w$ is constant and equal to $10^4$ mm$^{-1}$m$^{-3}$, only one $m$-$D$ relation representative of all BAECC cases, as presented in Section 4.3.1, is selected and $v$-$D$ representative of the snowfall with mean density ranging between 100 and 200 g cm$^{-3}$ is utilized. The $D_0$ values were varied between 0.5 and 4 and $\mu$ values between $-0.9$ and 3.

At the first stage of simulation a number of observed particles were computed assuming that it follows a Poisson distribution and the expected number of particles is determined by PIP sampling volume and the integration time that is determined by the precipitation accumulation. Given this number of particles, their diameters were found by sampling a Gamma probability density function, parameters of which are determined by the input PSD. To simulate the left-side truncation all particles with diameters smaller or equal to 0.25 mm, the PIP sensitivity threshold, were rejected. The right-side truncation was achieved by rejecting particles with sizes exceeding $3D_0$. For each $D_0$ and $\mu$ pair 50 simulated PSD were computed. Given the simulated truncated PSD the density is estimated in the same way as was presented above. This estimated density is compared to the one that is directly derived from the simulation input parameters and the results of their comparison is shown in Fig. 6. As one can see, the derived ensemble mean snow density is biased. The bias is largest for small $D_0$, which is explained by the left-side PSD truncation. For $D_0$ larger than 1 mm the bias decreases and approaches 2 %. Given that the error associated with PSD truncation is rather small, at least for $D_0$ larger than 1 mm and most of our observation fall within this range, in this study the truncation error is not corrected.

**3.5 Velocity-dimensional analysis**

For the retrieval of volume flux weighted snow density, velocity-dimensional relations of falling snow need to be estimated. For each integration time interval, $v(D) = a_v D^{b_v}$ is fitted for velocity-diameter data from the PIP. The $v(D)$ power law fits to unfiltered data tend to be strongly biased by outliers. To address this problem, Gaussian kernel density estimation (KDE, Silverman, 1986) is used to find the most probable velocity for each diameter bin, and only observations with velocities within half width at half maximum from the bin peak KDE value are included in calculating the fit. Using the linear least squares method, a fit is performed for the data points in log-log scale to derive a power law relation.  It should be noted, that using linear regression in log-log space does not optimally minimize residuals in linear space, but the method is used here as it does not overly emphasize the large end of the size spectrum. The retrieved velocity fits are shown for selected integration time intervals of the 18 March 2014 and the 22-23 January 2015 cases in the bottom of Figures 7 and 8, respectively.

It should be noted that the power law model, albeit widely used, may not necessary represent correctly velocities of ice particles over the complete range of diameters (Mitchell and Heymsfield, 2005). In many cases the fit can also be uncertain either because of narrow PSD or in presence of multiple particle types.

**4   Results**

**4.1   Case studies**

**4.1.1   18 March 2014**

During the March 18, Finland was covered in a continental polar air mass. In the morning, a warm occluded front associated with a weak low pressure center approached southern Finland from southwest bringing light snowfall. In the afternoon Hyytiälä was in the warm sector of the frontal system, and the relative humidity dropped halting the snowfall around 12 UTC. Later in the evening there was a one-hour snow shower from a squall line associated with a cold front passing over southern Finland.

Time series of LWE snow rate,  ensemble mean density and PSD parameters for the March 18 case are shown in Fig. 7. The bottom panels show measured fall velocities for selected integration time intervals representing observations with different  ensemble mean densities. Between the red dotted lines is the region where KDE is higher than half maximum for a given particle size. The fits are applied for data points between these lines. There is considerable scatter in particle fall velocity throughout the case and a bimodal PSD is present momentarily in the morning as can be seen in fall velocity panel Fig. 7a.

During the snow shower in the evening, liquid equivalent precipitation rates were recorded on average roughly three times more intense than earlier during the day , allowing retrievals of  $\bar{\rho}$ and PSD parameters at high time resolutions. Strong short time scale variations of  $\bar{\rho}$ and PSD parameters are recorded during this shower. The lowest  ensemble mean density value of the case, 0.035 $\mathrm{g\,cm^{-3}}$, is retrieved for time interval from 16:35 to 16:39, with concurrent $D_0$ value of 5.5 mm and $N_w$ of roughly 700 $\mathrm{mm^{-1}m^{-3}}$. The corresponding fall velocity distribution visualized in panel 7b is characterized by low values of velocity fit coefficients $a_v$ and $b_v$. Within the following 20 minutes, $D_0$ decreases down to roughly 2 mm, $N_w$ increases to $2 \times 10^4$ $\mathrm{mm^{-1}m^{-3}}$, and retrieved values of  $\bar{\rho}$ peak at over 0.2 $\mathrm{g\,cm^{-3}}$ between 16:54 and 16:58, and again from 17:05 to 17:08. Corresponding fall velocity distribution between 16:54 and 16:56, shown in panel 7c, is characterized by substantially higher values of $a_v$ and $b_v$, than 20 minutes earlier, which possibly indicates the onset of riming.

**4.1.2   22-23 January 2015**

During 22 January 2015, similarly to the 18 March 2014 event, a warm occluded front associated with a weak low moved northwards over the Gulf of Finland. However, due to a blocking high over north-western Russia, the low and the associated front were sustained over southern Finland for the whole day of January 23rd causing weak continuous precipitation in the area.

Time series of LWE snow rate,  $\bar{\rho}$ and PSD parameters for the 22-23 January 2015 case, with velocity-diameter fits from selected time intervals are shown in Fig. 8. The case is characterized by continuous snowfall at LWE precipitation rates lower than 1 mm h$^{-1}$ throughout the case. The velocity distribution for a given time interval has substantially less scatter compared to the 18 March 2014 case. The evolution of  $\bar{\rho}$ and $N_w$, as shown in Fig. 8, show considerable similarities, suggesting a strong correlation.

The velocity-diameter fits shown represent a low  ensemble mean density ($\bar{\rho} = 0.05\,\mathrm{g\,cm}^{-3}$) time interval 01:03-01:16 (panel 8b) and two intervals 22:30-22:52 and 02:06-02:14 (8a, c) with higher  values of $\bar{\rho}$, 0.10 and 0.12 g cm$^{-3}$, respectively. Notable is the higher modal fall velocities and the absence of particles larger than 3 mm in the high density time intervals compared to the distribution in panel 8b.

**4.2  v-D and density**

In Fig. 9, particle fall velocity versus diameter data points combined from all the cases in Table 1 are divided into three categories according to the  snow ensemble mean density of the time interval during which particles were observed. A least squares fit is applied to observations in each  $\bar{\rho}$ range using the same procedure as for velocity dimensional fits for integration time intervals, as described in section 3.5. The total number of observed particles is roughly 4,440,000, and for each  density category numbers of particles included in the fitting process (within the red lines in Fig. 9) are approximately 1,140,000, 1,190,000 and 360,000, respectively. The fitted relations for  ensemble mean density ranges are

$$v(D) = 0.834D^{0.217}, \quad 0.0\,\mathrm{g\,cm}^{-3} < \underline{\rho}\,\bar{\rho} \leq 0.1\,\mathrm{g\,cm}^{-3}, \tag{11}$$

$$v(D) = 0.895D^{0.244}, \quad 0.1\,\mathrm{g\,cm}^{-3} < \underline{\rho}\,\bar{\rho} \leq 0.2\,\mathrm{g\,cm}^{-3} \text{ and} \tag{12}$$

$$v(D) = 0.906D^{0.256}, \quad \underline{\rho}\,\bar{\rho} \geq 0.2\,\mathrm{g\,cm}^{-3}., \tag{13}$$

with RMSE values of 0.30 m s$^{-1}$, 0.30 m s$^{-1}$ and 0.35 m s$^{-1}$, respectively.

The coefficient is increased with density indicating higher fall velocities with more dense particles. There is also a clear increase in the slope of the fitted curve from the lowest  density range to the $0.1\ldots0.2$ g cm$^{-3}$ range indicated by the increase in the power term. With particles in the highest  density range the observed size distribution is narrow, hence the correlation between particle size and fall velocity is weak, and it is difficult to find an unambiguous relation between them. All things considered, the results are in line with the conclusion made by Barthazy and Schefold (2006), that the  prefactor and power terms increase with riming degree, which in turn are strongly connected with density (Power et al., 1964).

Considering the definition of the volume equivalent diameter, relations in the form of (11)...(13) should be ideal for velocity-dimensional parametrization of radar observations as the average size of hydrometeors as observed by radar are largely defined by their volumes rather than their shapes.

**4.3 Connection between PSD parameters and density**

From the analysis of PSD parameters and their relations to  ensemble mean density we have excluded data points representing integration time intervals where $D_0 < 0.6$ mm, as lower values of median volume diameter would imply that a substantial fraction of particles are too small to be observed with PIP. Applying this restriction, along with minimum thresholds set for particle count and LWE precipitation rate in density retrievals, as described in section 3.2, all in all 101 time intervals were discarded from the total of 1141 intervals of observations, leaving 7173 minutes of snow observations for the analysis.

**4.3.1 Density and $D_0$**

In Fig. 10, observed distributions of $D_0$ for the three different density regimes are shown. For the low density particles, the maximum $D_0$ value  rarely exceeds 5 mm, which is in agreement with observations of snow aggregates presented by Lo and Passarelli (1982). It can also be seen that $D_0$ distribution depends on density. Low density particles are generally larger and vice versa. This dependence of $D_0$ on  ensemble mean density is not surprising, given that they are related as was previously shown by B07 and discussed in more detail below.

Relation between  $\bar{\rho}$ and size ($D_0$) is illustrated in Fig. 11. The areas of individual data points are proportional to the particle counts of the corresponding observation time intervals. The overlaid black solid curve, a least squares fit applied for all cases in Table 1 is given by

$$\rho\bar{\rho}(D_0) = 0.226 D_0^{-1.004}, \tag{14}$$

where $D_0$ is in millimeters and $\rho\bar{\rho}$ is in g cm$^{-3}$. As the two examined winters were seen to have notable differences between each other in the snowfall type and average  $\bar{\rho}$, corresponding relations were also calculated separately for the winters, and are given by

$$\rho\bar{\rho}(D_0) = 0.273 D_0^{-0.998} \quad \text{and} \tag{15}$$

$$\rho\bar{\rho}(D_0) = 0.209 D_0^{-0.969} \tag{16}$$

for BAECC events and for events of winter 2014/15, respectively. A relation by B07, given by  $\bar{\rho}(D) = 0.178 D_0^{-0.922}$, is plotted in Fig. 11 for comparison. As their definitions of particle diameter and  $\bar{\rho}$ are close to ours, the relations are easy to compare. Especially (16) is in good agreement with B07's results. The  ensemble mean density is on average higher for snow events recorded during BAECC, which suggests more riming occurred during those events. Indication to this is that the ARM AMF2 dual-channel microwave radiometer located on the same measurement field detected the presence of liquid water more than 80 % of the BAECC SNEX campaign time (Petäjä et al., 2016) and the presence of supercooled liquid layers could also be observed in the backscatter coefficient and circular depolarization ratio measurements of the co-located ARM HSRL (High Spectral Resolution Lidar) in the majority of the BAECC cases (Goldsmith et al., 2014). In general the BAECC winter was milder than the next winter 2014–2015, and the case duration weighted average of maximum recorded

temperatures was almost one degree higher for BAECC events compared to the value for winter 2014–2015 cases. The temperatures closer to 0°C could mean increased aggregation as stated in B07 and therefore decreased density values, but also different snow habits compared to  colder cases.

The mass-dimensional relation in power-law format  $m = a_m D^{b_m}$ can be derived from the retrieved  $\bar{\rho}$-$D_0$ relations (14) to (16) by assuming  gamma PSD and describing the  ensemble mean density approximately as

$$\rho\ \bar{\rho} = \frac{\int m(D)v(D)N(D)\mathrm{d}D}{\int V(D)v(D)N(D)\mathrm{d}D} \approx \frac{\int_0^\infty m(D)v(D)N(D)\mathrm{d}D}{\int_0^\infty V(D)v(D)N(D)\mathrm{d}D} \tag{17}$$

$$= \frac{\int a_m (D)^{b_m} a_v D^{b_v} N_0 \exp(-\Lambda D)\mathrm{d}D}{\int \frac{\pi}{6}(0.1D)^3 a_v D^{b_v} N_0 \exp(-\Lambda D)\mathrm{d}D}\ \frac{\int_0^\infty a_m (D)^{b_m} a_v D^{b_v} N_0 D^\mu \exp(-\Lambda D)\mathrm{d}D}{\int_0^\infty \frac{\pi}{6}(0.1D)^3 a_v D^{b_v} N_0 D^\mu \exp(-\Lambda D)\mathrm{d}D} \tag{18}$$

$$= \frac{6}{\pi}\ \frac{6}{\pi} 10^3 a_m \frac{\Gamma(b_m + b_v + 1)}{\Gamma(4 + b_v)} D_0^{b_m - 3}\ \frac{\Gamma(b_m + b_v + \mu + 1)}{\Gamma(b_v + \mu + 4)} \left(\frac{0.1}{3.67}\ \frac{1}{3.67}\right)^{b_m - 3} D_0^{b_m - 3}. \tag{19}$$

The integration limits are defined from zero to infinity for deriving the analytic solution, though the true range is narrower because of left and right truncation of the observed size spectrum. As shown in Fig. 6, the ensemble mean density is overestimated because of the truncation. The estimation bias of density is ranging between 20 % for $D_0$ smaller than 0.75 mm and about 2 % for $D_0$ larger than 2 mm. Since for the estimation of the $m$-$D$ relation, most of the observed $D_0$ values are higher than approx. 1 mm as shown in Fig. 10, there is only minor contribution of the smaller $D_0$ values, and we are assuming our error in ensemble mean density because of truncation to be close to 2 %. This corresponds to an error of 2 % also in the prefactor $a_m$, if assumed that the truncation does not introduce significant changes in the exponents of the $\bar{\rho}$-$D_0$ and $m$-$D$ relations.

Taking the three velocity exponents from equations (11) to (13), and assuming exponential PSD, the derived prefactors and exponents of mass-relation  are shown in Table 2, having the volume-equivalent diameter proxy in mm and mass given in grams. The factor 0.1 in (18) derives from unit conversion, as  $\bar{\rho}$ is in g cm$^{-3}$. The values of prefactor $a_m$ are not sensitive to the changes in the velocity exponent $b_v$ (changes in $b_v$ are resulting less than 1 % devition $a_m$ values), though there is a small increase in $a_m$ with increasing $b_v$. The prefactor is more sensitive to shape parameter $\mu$ of the gamma PSD, the value of $a_m$ increases by 24 % as $\mu$ is increased from zero to 1. With value of $\mu = 3$ the increase in the prefactor $a_m$ value is 48 %. The shape factor of snow PSD is known to be noisy and thus often exponential distribution is assumed. With $b_v = 0.217$ the derived mass-dimensional relations for all cases and for both studied winters separately are plotted against literature values in Fig. 12. The derived exponent $b_m$ for the studied cases is in line with literature values, close to 2, but the prefactor $a_m$ values are higher than  the presented relations in Table 3. The highest value of $a_m$ is for the BAECC cases indicating conditions of riming. The high prefactor values might manifest the Finnish winter conditions, because of the vicinity of Baltic Sea, the air is more moist than e.g. in continental conditions.

**4.3.2  $N_w$ and density**

Distributions of observed $N_w$ values also exhibit dependence of $N_w$ on the  ensemble mean density, as shown in Fig. 13, i.e. $N_w$ increases with density. The modal values of $N_w$ are approximately 5000, 40,000 and 80,000 mm$^{-1}$m$^{-3}$ for  ensemble mean density ranges $0.0\dots0.1$, $0.1\dots0.2$ and $>0.2$ g cm$^{-3}$, respectively, with vast majority of $N_w$ values spanning less than two orders of magnitude for a given  $\bar{\rho}$ range. This dependence of $N_w$ on density is somewhat unexpected. There is no obvious reason to expect that $N_w$ would depend on density. However, because $D_0$ and density are related, dependence of $N_w$ on density potentially arises from the dependence of $N_w$ on $D_0$.

To verify this, the partial correlation analysis of the relation between log values of $N_w$ and density while controlling for log value of $D_0$ was carried out. It was found that there is a moderate negative partial correlation, -0.33, between $N_w$ and density while controlling for $D_0$. However, the zero-order correlation between $N_w$ and density is 0.52. The analysis confirms that the observed relation between $N_w$ and density, is due to their relation to $D_0$. It is not clear, however, what is the meaning of the found negative partial correlation between $N_w$ and density.

[revised manuscript text omitted]

*Acknowledgements.*   We would like to acknowledge the Hyytiälä station and University of Helsinki personnel for the daily tasks with measurements, especially mentioning Matti Leskinen and Janne Levula. The research of JT and DM was supported by Academy of Finland (grant 263333) and the Academy of Finland Finnish Center of Excellence program (grant 272041). AvL was funded by grant of the Vilho, Yrjö and Kalle Väisälä Foundation and by SESAR Joint Undertaking Horizon 2020 grant agreement No 699221 (PNOWWA). The instrumentation used in this study was supported by NASA Global Precipitation Measurement Mission ground validation program and by the Office of Science U.S. Department of Energy ARM program.

[Figure]

**Figure 1.** Snow precipitation instruments on the measurement field in Hyytiälä. The Pluvio² 200 is inside the wind protection on a platform and the PIP lamp can be seen at right on the ground. The view of the picture is to southwest and the distance from the platform to the treeline behind is approximately 20 m.

[Figure]

**Figure 2.**  ratio of the diameter observed by PIP, $D_{PIP}$, to volume equivalent diameter $D$.

[Figure]

**Figure 3.** The distributions of snowflake a) aspect ratio and b) area ratio as observed using PIP with interquartile ranges visualized and median values shown.

[Figure]

**Figure 4.** Distribution of snow ratios, ratio of snow depth change to LWE, calculated from retrieved ensemble mean densities with interquartile range, and median and mean values.

[Figure]

**Figure 5.** Scatterplot of the hourly change of snow depth measured with Jenoptik SMH30 and estimated from volume flux using PSD and fall velocities as measured by PIP. The data includes all the studied cases except Jan 10-11 2015.

[Figure]

**Figure 6.** Computed normalized bias and standard deviation of estimated mean snow density as a function of $\mu$ and $D_0$. The shaded area indicates data that is not included in the analysis, because derived $D_0$ is smaller than 0.6 mm. The increased values of bias at low $D_0$ values is due to left side truncation of the observed PSD, which is caused by the instrument sensitivity. At larger $D_0$ values the bias approaches value of 0.02.

[Figure]

**Figure 7.** Evolution of snowfall intensity,  ensemble mean density and particle size distribution parameters during March 18th 2015 with associated $(v, D)$ from three selected time intervals (highlighted in gray). The red dashed lines mark the upper and lower velocity limits where for a given $D$, the KDE value is higher than half maximum.

[Figure]

**Figure 8.** Evolution of snowfall intensity,  ensemble mean density and particle size distribution parameters during the night between the 22nd and 23rd of January 2015 with associated $(v, D)$ from three selected time intervals (highlighted in grey). The red dashed lines mark the upper and lower velocity limits where for a given $D$, the KDE value is higher than half maximum.

[Figure]

**Figure 9.**  Probability densities of $(D, v)$ in three  ensemble mean density ranges ($[\bar{\rho}] = \mathrm{g\,cm}^{-3}$). Dashed lines mark the full width at half maximum KDE in each diameter bin. Power law functions are fitted for data between those lines.

[Figure]

**Figure 10.** Normalized frequency (bars) and kernel density (line) of median volume diameter $D_0$ in three  ensemble mean density ranges,  $[\bar{\rho}]$ = g cm  $^{-3}$.

[Figure]

**Figure 11.** ($D_0$, $\bar{\rho}$) for all cases listed in  Table 1. Area of each dot is proportional to the number of particles in corresponding integration time interval. Power law fits are shown separately for BAECC winter cases (blue) and cases from the following winter (green).

[Figure]

**Figure 12.** Derived $m$-$D$ relations assuming exponential PSD in comparison relations presented in literature are shown in Table 3.  A conversion of maximum dimension to volume equivalent diameter is done by assuming axis ratio of 0.6.

[Figure]

**Figure 13.** Frequency of $N_w$ in three  ensemble mean density ranges,  $[\bar{\rho}]$ = g cm$^{-3}$.

[Figure]

**Figure 14.** $(D_0, N_w)$ and ($\sout{D_0\rho^{1/3}}\underwave{D_0\bar{\rho}^{1/3}}$, $N_w$) with fitted relations

[Figure]

**Figure 15.** Normalized frequency (bars) and kernel density (line) of the gamma PSD shape factor $\mu$ in three  ensemble mean density ranges, $[\rho]$ $[\bar{\rho}]$ = g cm$^{-3}$ $^{-3}$.

**Table 1.** Liquid water equivalent precipitation accumulation measured with Pluvio[2] 200 and 400, change in snow depth and maximum and minimum temperature,maximum and minimum relative humidity, mean and maximum wind speed and mean wind direction of the studied snow events. Events before the horizontal line are recorded during the BAECC campaign.

| Event | LWE (mm) | | $\Delta$SD | Temp (°C) | | RH (%) | | Wind (m s$^{-1}$, °) | | |
| --- | --- | --- | --- | --- | --- | --- | --- | --- | --- | --- |
| | 200 | 400 | (cm) | min | max | min | max | mean | max | mean dir. |
| 2014 Jan 31 21:00 - Feb 01 06:00 | 7.4 | 7.3 | 5.1 | -9.8 | -8.9 | 84 | 91 | 1.6 | 2.9 | 138 |
| 2014 Feb 12 04:00 - 11:00 | 1.0 | 0.9 | 1.8 | - 1 | 0 | 96 | 98 | 0.6 | 2.0 | 170 |
| 2014 Feb 15 21:00 - Feb 16 03:00 | 2.6 | 2.6 | 2.5 | -2.1 | -1 | 86 | 97 | 1.9 | 2.7 | 140 |
| 2014 Feb 21 16:00 - Feb 22 05:00 | 5.5 | 5.2 | 3.6 | -2.7 | 0 | 88 | 98 | 2.1 | 3.4 | 138 |
| 2014 Mar 18 08:00 - 19:00 | 4.4 | 4.0 | 7.3 | - 3.8 | -1.8 | 76 | 96 | 1.2 | 2.7 | 155 |
| 2014 Mar 20 16:00 - 23:00 | 6.1 | 5.9 | 4.8 | - 4.3 | -1.3 | 89 | 97 | 2.0 | 3.4 | 146 |
| 2014 Nov 06 19:00 - Nov 07 14:30 | 10.5 | – | 10.3 | -2.4 | -1.6 | 95 | 97 | 0.8 | 1.9 | 238 |
| 2014 Dec 18 14:00 - 19:00 | 2.6 | 2.2 | 3.9 | -2.3 | -0.8 | 97 | 98 | 1.0 | 1.8 | 134 |
| 2014 Dec 24 08:30 - 13:00 | 1.3 | 1.2 | 1.2 | -9.2 | -8.9 | 90 | 91 | 0.7 | 1.5 | 204 |
| 2014 Dec 30 00:30 - 14:00 | 6.3 | 5.3 | 4.9 | -10.4 | -0.6 | 91 | 98 | – | – | – |
| 2015 Jan 3 09:00 - 23:50 | 7.3 | 7.3 | 11.9 | - 3.9 | 0 | 96 | 98 | 2.6 | 5.2 | 318 |
| 2015 Jan 7 01:00 - 20:10 | 5.4 | 4.8 | 2.2 | -6.5 | -0.8 | 92 | 97 | 1.3 | 2.8 | 181 |
| 2015 Jan 8 06:00 - 13:30 | 2.6 | 2.7 | 1.6 | - 1.9 | 0 | 97 | 99 | 1.0 | 2.2 | 155 |
| 2015 Jan 9 18:00 - Jan 10 06:00 | 3.1 | 3.1 | 4.6 | -3.7 | -0.2 | 95 | 98 | 1.0 | 3.0 | 286 |
| 2015 Jan 10 22:00 - Jan 11 09:00 | 0.7 | 0.6 | 0.7 | -12.6 | -4.4 | 88 | 95 | 1.6 | 3.4 | 207 |
| 2015 Jan 12 21:00 - Jan 13 08:30 | 12.8 | 10.9 | 9.6 | -15.7 | -9.0 | 88 | 94 | 1.3 | 3.1 | 181 |
| 2015 Jan 13 22:00 - Jan 14 07:00 | –* | 2.2 | 1.9 | -8.0 | -0.3 | 94 | 98 | 0.5 | 1.9 | 134 |
| 2015 Jan 16 01:30 - 07:30 | –* | 5.8 | 5.2 | -1.3 | -0.6 | 92 | 98 | 1.9 | 3.4 | 154 |
| 2015 Jan 18 16:00 - 21:00 | 1.9 | 1.9 | 2.7 | -2.4 | -0.3 | 95 | 97 | 1.2 | 2.6 | 300 |
| 2015 Jan 22 21:00 - Jan 23 04:30 | 2.1 | 2.0 | 2.3 | -13.3 | -12.5 | 87 | 90 | – | – | – |
| 2015 Jan 23 15:00 - 23:00 | 1.4 | 1.2 | 1.4 | -10.1 | -8.8 | 91 | 93 | 0.3 | 1.0 | 205 |
| 2015 Jan 25 09:00 - 16:00 | 2.8 | 2.5 | 1.9 | -2.4 | -1.7 | 96 | 97 | 0.7 | 1.7 | 170 |
| 2015 Jan 31 12:00 - Jan 31 23:15 | 7.0 | 6.6 | 5.7 | -1.9 | -0.4 | 92 | 97 | 1.2 | 2.6 | 175 |

*Pluvio[2] 400 was used as data from Pluvio[2] 200 was unavailable

**Table 2.** The prefactors and exponents of $m = a_m D^{b_m}$ derived for exponential PSD with different values of exponent $b_v$ of velocity relation. The mass given in grams and the volume-equivalent diameter proxy in mm.

| Dataset | $b_m$ | $a_m(b_v = 0.217)$ | $a_m(b_v = 0.244)$ | $a_m(b_v = 0.256)$ |
|---|---|---|---|---|
| All cases | 1.996 | 1.036e-4 | 1.045e-4 | 1.049e-4 |
| BAECC cases | 2.002 | 1.254e-4 | 1.264e-4 | 1.269e-4 |
| Winter 2014-2015 cases | 2.031 | 9.679e-5 | 9.757e-5 | 9.792e-5 |

**Table 3.** The prefactors and exponents of $m = a_m D^{b_m}$ of literature values for comparison plotted in Fig. 12. A conversion from maximum dimension to volume equivalent diameter is done by assuming axis ratio of 0.6.

| Study | $b_m$ | $a_m$ |
|---|---|---|
| Matrosov 2007 , $0.12\,\mathrm{mm} < D \leq 2.4\,\mathrm{mm}$ | 2.0 | $4.2172 \times 10^{-5}$ |
| Matrosov 2007 , $2.4\,\mathrm{mm} < D \leq 24\,\mathrm{mm}$ | 2.5 | $3.2430 \times 10^{-5}$ |
| Heymsfield et al. 2004 | 2.04 | $7.5814 \times 10^{-5}$ |
| Mitchell et al. 1990 | 2.0 | $3.0926 \times 10^{-5}$ |
| Locatelli and Hobbs 1974 | 1.9 | $5.1134 \times 10^{-5}$ |

---

## Author Comment (AC5) · 26 Aug 2016

A comment meant to be posted in response to the Anonymous Reviewer #3 was accidentally posted in the comment thread of reviewer A. Heymsfield. We apologize this human error. The responses are now also posted in their correct threads.
* * *